# Systematically identification of survival-associated eQTLs in a Japanese kidney cancer cohort

Xiya Song[1], Han Jin[1], Xiangyu Li[1], Meng Yuan[1], Hong Yang[1], Yusuke Sato[2], Haruki Kume[2], Seishi Ogawa[3], Cheng Zhang[1,4]*, Adil Mardinoglu[1,5]*

**1** Science for Life Laboratory, KTH - Royal Institute of Technology, Stockholm, Sweden, **2** Department of Urology, Graduate School of Medicine, The University of Tokyo, Tokyo, Japan, **3** Department of Pathology and Tumor Biology, Institute for the Advanced Study of Human Biology (WPI-ASHBi), Kyoto University, Kyoto, Japan, **4** The Roger Williams Institute of Liver Studies, Faculty of Life Sciences & Medicine, King's College London, London, United Kingdom, **5** Centre for Host-Microbiome Interactions, Faculty of Dentistry, Oral & Craniofacial Sciences, King's College London, London, United Kingdom

* cheng.zhang@scilifelab.se (CZ); adilm@scilifelab.se (AM)

## Abstract

### Background

Clear cell renal carcinoma (ccRCC) is the predominant form of kidney cancer, but the prognostic value of expression quantitative trait loci (eQTLs) remains underexplored, particularly in Asian populations.

### Objective

We analyzed whole-exome sequencing and RNA sequencing data from 100 Japanese ccRCC patients to identify eQTLs. Multiple Cox proportional hazard models assessed survival associations, with validation in the Cancer Genome Atlas ccRCC cohort (n = 287).

### Results

We identified 805 eGenes and 4,558 cis-eQTLs in the Japanese cohort. Survival analysis revealed a total of 9 eGenes significantly associated with overall survival (FDR < 0.05). Further exploratory analysis were performed using 158 eGenes and 711 eQTLs (p-value <0.05) as potential prognostic signals. Among these, 223 eQTLs regulating 54 eGenes showed consistent prognostic effects at both expression and genetic levels. Cross-population validation identified eight eQTLs regulating 11 eGenes with reproducible survival associations across ethnicities, including a missense mutation in ERV3–1 and regulatory variants near ANKRD20A7P. These variants demonstrated consistent allelic effects on both gene expression and patient survival in both cohorts.

**Data availability statement:** The Japanese cohort dataset from the European Genome-phenome Archive (https://www.ebi.ac.uk/ega/studies/EGAS00001000509) and the TCGA KIRC dataset (https://portal.gdc.cancer.gov/projects/TCGA-KIRC) are available upon application of access to the original data generation facilities. The full summary statistics of the eQTL analysis on the JP cohort (n=100) are available on Zenodo (DOI No. 10.5281/zenodo.15333222, https://doi.org/10.5281/zenodo.15333223), with full public accessibility. Code for eQTL analysis is available in: https://github.com/xiyasong/eQTL-analysis; Code for survival analysis is available in: https://github.com/xiyasong/eQTL_manuscript.

**Funding:** The study is funded by the Knut and Alice Wallenberg Foundation, Sweden, under project number 72110 (to A.M.). X.S, H.J, X.L,M.Y and H.Y received salary from 72110. The funders had no role in study design, data collection and analysis, decision to publish, or preparation of the manuscript.

**Competing interests:** I have read the journal's policy and the authors of this manuscript have the following competing interests: A.M is the co-founder of SZA Longevity Inc, San Diego, CA, USA. The company was not involved in the design, funding, execution, or publication of this research. The remaining authors declare no competing interests.

## Author summary

Clear cell renal carcinoma (ccRCC) is the most common type of kidney cancer, but effective targeted therapies and reliable tools to predict patient outcomes are still limited. In this study, we explored how inherited genetic differences influence gene activity and survival in patients with ccRCC. Specifically, we focused on genetic variants known as expression quantitative trait loci (eQTLs), which affect how much certain genes are expressed. Our research is the first to examine the prognostic value of these eQTLs in an Asian patient population. By analyzing genomic and gene expression data from Japanese patients, and then validating our findings in an independent international cohort from The Cancer Genome Atlas (TCGA), we identified several genetic variants that are consistently associated with patient prognosis across populations. Notably, we discovered a variant in the ERV3–1 gene that alters the protein structure, as well as regulatory variants near genes like ANKRD20A7P that may influence cancer progression. These findings provide new insights into the genetic architecture of ccRCC and suggest potential biomarkers that warrant further investigation.

## 1. Introduction

Clear cell renal cell carcinoma (ccRCC) is the most common subtype of renal cell carcinoma (RCC), accounting for the majority of kidney cancer-related deaths [1]. ccRCC has a strong inheritable genetic predisposition [2] and is recognized as a highly polygenic and complex disease. Genetic mutations, especially in VHL, PBRM1, SETD2, and BAP1 play a crucial role in ccRCC [3]. The highly heterogeneous genomic profiles of ccRCC also lead to various survival outcomes among patients [1,4,5]. Therefore, identifying the genetic risk loci for ccRCC is crucial to understanding its mechanisms, predicting the prognosis and developing new targeted treatments.

Although genome-wide association studies (GWAS) were widely employed in discovering genetic risk factors for ccRCC [6,7], many of the risk SNPs identified reside in non-coding regions [8] which brings difficulties in interpreting the biological functions and applying them to clinics. Expression quantitative trait loci (eQTL) analysis can link genetic loci to gene expression, uncovering "eGenes" that are influenced by these loci. These eGenes are valuable for functional follow-up studies and have been reported as candidate susceptibility genes in cancer [9], potential cancer driver genes [10], and survival-associated genes in various cancers [11]. Moreover, these genes with genetic associations are also more likely to serve as drug targets [12].

Several studies have established public databases for kidney-specific eQTLs. The GTEx Consortium [13] identified eQTLs from 73 healthy kidney samples, while ccRCC-specific eQTL analyses have predominantly been conducted using data from The Cancer Genome Atlas Kidney Renal Clear Cell Carcinoma (TCGA-KIRC). For example, Gong et al. investigated ccRCC cancer eQTLs by analyzing germline variants and tumor transcriptomics [14], while Yang et al. explored somatic mutations

and their associations with gene modules [15]. However, both the GTEx and TCGA datasets have a limited representation of Asian populations (1.3% in GTEx, 3% in TCGA) [16]. Given that gene variations can be ethnically distinctive, these findings lack the translation into a broad sense. Further identification of eQTLs in ccRCC is required for patients from diverse ethnic backgrounds including Asian populations. Furthermore, the impact of germline variants with significant eQTL signals on prognosis remains understudied in ccRCC. While some common germline variants have been linked to ccRCC risk and outcomes, significant eQTL signals in these variants have yet to be reported [17–19]. In other cancers, eQTLs of prognostic genes have been linked to overall survival, such as in multiple myeloma [20]. To date, there are no studies to date focusing on survival-associated eQTLs and eGenes in ccRCC within Asian populations.

In this study, we conducted a comprehensive eQTL analysis on a Japanese ccRCC cohort (JP cohort). We first compared our results with existing kidney and ccRCC-specific databases, including GTEx release version 8 (v8) and a Pan-cancer eQTL database, to identify the shared ccRCC eGenes/eQTLs across all three cohorts and the eQTLs unique to the JP cohort. We then performed survival analysis using Cox proportional hazard models to identify prognostic eQTLs that were either unique to the JP cohort or conserved across diverse ethnic backgrounds. Different inheritance models for SNPs were considered in the survival analysis, and we controlled for known clinical parameters. To establish the robustness of our findings, we examined the pairwise prognostic effects by assessing the consensus impact of eQTL-eGene pairs. The prognostic effects were visualized using Kaplan-Meier curves and forest plots. Finally, we validated significant prognostic findings from the JP cohort using genotype and survival data from The Cancer Genome Atlas (TCGA) ccRCC cohort. The study design is illustrated in Fig 1.

## 2. Materials and methods

### 2.1. Ethics statement

The JP cohort data were obtained from the published study with data access request to the corresponding authors [21], in which the original sample collection was approved by the Graduate School of Medicine at the University of Tokyo's ethics committee. A written informed consent was obtained along with ethics approval for the molecular analysis.

A formal application was submitted through the TCGA portal to access controlled data through the TCGA whole-exome sequencing (WES) dataset. The Cancer Genome Atlas (TCGA) collected both tumor and non-tumor biospecimens from human samples with written informed consent under the authorization of local institutional review boards (https://www.cancer.gov/ccg/research/genome-sequencing/tcga/history/ethics-policies). Additional ethical approval was not required in this study.

### 2.2. Study subjects

The eQTL analysis was conducted on 100 Japanese ccRCC patients (JP cohort) with tumour whole exome sequencing (WES) and transcriptome sequencing (RNA-seq) data obtained from the previous study [21]. The cohort consists of 23 females and 77 males (aged from 35 to 91). The clinical information for the JP cohort was retrieved from the metadata. The living days were calculated by the last follow-up to alive patients and days to death to dead patients, which ranged between 30 days to 4350 days. All the sequence data were downloaded from the European Genome-Phenome archive in the BAM format (Accession number: EGAD00001000597).

A total of 287 ccRCC WES data (TCGA cohort) that were paired-end sequenced among all read groups from the TCGA-KIRC cohort were included. The corresponding patients' clinical data were downloaded by using the R package TCGAbiolinks [22]. This dataset was included as a validation cohort for the identified survival-associated eQTLs. The chosen TCGA cohort consists of 102 females and 185 males. The living days were calculated as the same as the JP cohort, which ranges between 3 days to 4537 days.

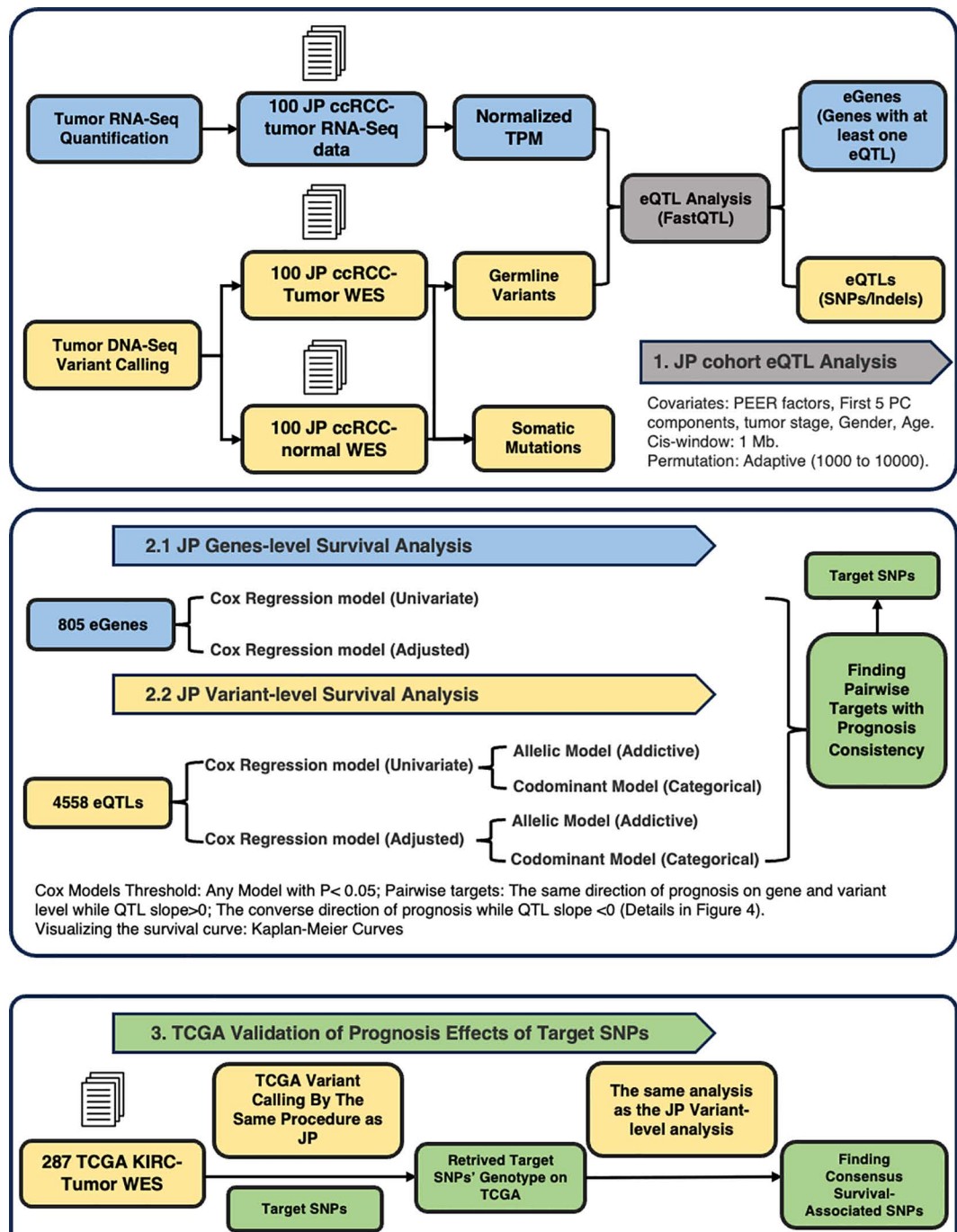

**Fig 1. The general study design and workflow of the whole analysis.** This workflow illustrates the integrative analysis conducted on Japanese (JP) ccRCC and TCGA KIRC datasets to identify survival-associated genetic variants. In the JP cohort, RNA-Seq and WES data from 100 ccRCC tumors were used to perform eQTL analysis (FastQTL), identifying eGenes and eQTLs, with covariates such as tumor stage, age, and gender. Cox regression models assessed survival impact at both gene (805 eGenes) and variant (4558 eQTLs) levels, exploring allelic and codominant models and identifying prognosis-consistent targets. Validation in the TCGA KIRC dataset used 287 tumor samples to replicate JP findings, focusing on consensus survival-associated SNPs across cohorts.

The patient's detailed clinical characteristics of the JP and the TCGA cohort included in this study are presented in S1 Fig. This includes age, gender, living days, stage at diagnosis (T), lymph node spread (N), and metastases stage (M).

## 2.3. Data pre-processing

The tumor gene expression profiles of the JP cohort were retrieved following the steps outlined in the GTEx consortium's public GitHub repository (https://github.com/broadinstitute/gtex−pipeline). The RNA-seq of the 100 ccRCC tumor tissues were mapped to the reference genome GRCh38/hg38 with the annotation file GENCODE v26(https://www.gencodegenes.org/human/release_26.html). The BAM files were transformed to paired FASTQ files by the Picard toolbox (https://broadinstitute.github.io/picard/). Next, the FASTQ files were aligned using STAR (v2.5.3a) with two-pass mode [23]. The duplicates were removed by Picard. Finally, the gene-level quantification was calculated by RNA-SeQC (v2.3.6) [24]. During the process, the read-level filters were applied by RNA-SeQC to ensure the high quality of gene abundance quantification.

The variants calling, quality control and filtration were processed based on The Genome Analysis Toolkit (GATK) best practice 4.1.1.0 [25] and the workflow of GTEx version 8(GTEx v8). To utilize the WES data to explore regulatory effects on non-coding regions, the variant calls were kept if passing quality control, regardless in WES-exon target regions or in off-target regions [26]. First, the BAM files were all converted into FASTQ files and re-mapped to the reference GRCh38 using BWA-MEM [27]. Second, the GATK IndelRealigner was applied for local realignment. Then, the duplicate removal and base quality score recalibration (BQSR) were performed by MarkDuplicates and BaseRecalibrator. Finally, GATK HaplotypeCaller, GenomicsDBImport and GenotypeGVCFs by GVCF mode were applied sequentially to perform the joint-calling of variants (SNVs, Indels) on the whole JP cohort. Similar to the GTEx v8, the variants calling was restricted to autosomes and chromosome X.

The variants generated by joint calling were filtered out based on multiple quality control (QC) steps and also to remove the rare variants for preparation of eQTL analysis. First, the variants with the value of Excess Heterozygosity (Excess-Het) > 54.69 were removed before the Variant Quality Score Recalibration (VQSR). Second, we implemented the VQSR with allele-specific mode to exclude the SNPs with a threshold of 99.8% and Indels with the threshold of 99.95%. The variants with QUAL <30, QualByDepth (QD) <2.0 or Inbreed Coefficient <-0.3 were excluded to ensure that we filtered low-confidence variants, including those with insufficient read depth and with excess heterozygosity. The multi-allelic sites were split into biallelic sites using Hail 0.2 [28]. Next, monomorphic SNPs without called alternative alleles (AC==0) were removed using bcftools (https://samtools.github.io/bcftools). We removed the variants with genotype missingness <25% at each site, as well as variants with minor allele frequencies (MAF) <= 0.01. The remaining variants were included in the downstream eQTL analysis (S1 Table).

To increase the quality of genotype calls, we applied genotype-level filtering through the GATK genotype refinement pipeline to re-calculate the GQ value per sample at per variant site. The genotypes with re-calculated GQ < 20 or the minimum read depth (DP) <3 were trimmed off by GATK VariantFiltration. After the filtration, the total number of variant were not changed but the unqualified genotypes were all set to missing genotypes.

The RS IDs of the variants identified in the WES samples of the Japanese cohort were retrieved using dbsnp version 138 by the GATK variant annotator. To better represent the variants and compare them with GTEx results, we concatenated the variants' chromosome, location, reference allele and alternative allele to create unique IDs. To analyze the variant types and function classes, we used SnpEff [29].

Lastly, the germline variants calling of the TCGA cohort was performed following the GATK best practice, while the software version, parameters, and the procedures strictly followed the steps used for processing the JP cohort to ensure comparability.

## 2.4. Somatic mutation analysis on the JP cohort

The GATK Mutect2 was applied to the prepared tumor BAM files and matched normal BAM files of the JP cohort to identify somatic variants. While using the'Tumor with matched normal' model, a panel of normal (PON) files provided by GATK

was also used to filter the germline variants. FilterMutectCalls was used to filter the somatic mutations which considered read orientation bias and cross-contamination.

## 2.5. eQTL analysis on the JP cohort

The Docker image provided by GTEx v8 was used for running eQTL analysis by the developed software FastQTL [30]. The gene-level expression data from 100 samples were normalized based on the QTL pipeline in the GTEx Consortium GitHub repository. Specifically, the read counts were normalized between samples using TMM [31]. Then, Genes were selected based on expression thresholds of >0.1 TPM in at least 20% of samples and ≥6 reads in at least 20% of samples. The nominal p-value was calculated by FastQTL.

We included multiple factors as covariates in eQTL analysis of the JP cohort. First, the probabilistic estimation of expression residuals (PEER) [32] was used and the number of PEER factors (15) was set based on the sample size. Second, the first 5 PCs were calculated using PLINKv1.9 [33] and further included as covariates. Additionally, the gender (male/female) and age of patients (below 50 years old/above 50 years old) were also included. Finally, the patient's stage at diagnosis was also included as one of the covariates. Specifically, the tumour size (T) was encoded as an integer between 1–4. Whether Cancer has spread into nodes(N) or not was encoded as integer 0,1 or 2 (represents the extent of the spread). The metastasis situation (M) of the patients was encoded as 0 or 1, depending on whether the cancer had metastasized to distant tissues. The linear regression models for each testing pair were built as the following formula:

$$Y_i = \beta_0 + \beta_1 x_{i1} + \beta_2 x_{i2} + ... + \beta_p x_{ip} + \epsilon_i$$

Where $Y_i$ represents the gene expression, and $x_i$ represents all the factors that might regulate the gene expression, including genotypes and covariates of each sample as we mentioned above.

To calculate the nominal p-value, the cis-mapping window was set to 1Megabase (Mb) between variants and the transcription start site (TSS) of genes. The variant IDs were named using variant information (chromosome, position, REF, ALT) to keep the same format as the GTEx. An adaptive permutation pass between 1000–10000 times was used to calculate the adjusted p-value for each gene's top associated variant. The missing genotypes were imputed to mean dosage across samples by FastQTL within the regression model. No external genotype imputation was performed. Extra gene and variant information were added to the final eQTL results table. To generate a list of all the pairs of significant cis-eQTL and genes, we applied an FDR threshold of <0.05 to all the nominal tested pairs.

To further assess the robustness of top associations and minimize potential bias due to low sample counts, we performed additional sensitivity analyses using addictive, dominant and recessive genetic models. For each model, we performed simple linear regression (lm(expression~genotype)) to evaluate association significance and consistency across models. Top findings that failed to reach significance in all tested models and had fewer than three samples in both the heterozygous and homozygous alternative genotype groups were excluded from further analysis.

The classification and function annotation of identified eGenes and eQTLs were acquired by the Ensembl BioMart and Ensembl Variant Effect Predictor (VEP).

## 2.6. Comparison of eQTL analysis with existing eQTL databases

To compare our cis-eQTL results with the established databases, the public eQTL datasets were applied from: GTEx v8, which comprises 73 kidney cortex samples from healthy individuals, and the Pan-cancer database includes 527 ccRCC tumour samples from The Cancer Genome Atlas (TCGA) [14]. The eGenes and eQTLs identified from these samples were retrieved from Kidney_Cortex.v8.egenes.txt with q value <0.05 and Kidney_Cortex.v8.signif_variant_gene_pairs.txt for the GTEx v8 database and KIRC_tumor.cis_eQTL.txt for the Pan-cancer database.

## 2.7. Regulatory potential analysis

To explore the regulatory functions of selected eQTLs, we used LDassoc [34] module to calculate their Linkage disequilibrium (LD) using the population of Japanese in Tokyo (JPT). Then, we labelled the level of regulatory functionalities and retrieved the chromatin state of the specific eQTL by RegulomeDb v2.0.3 [35].

## 2.8. Survival analysis on JP and TCGA cohorts

Univariate survival analysis by the Cox proportional hazards regression model (Cox) was performed in the JP cohorts, using the same TPM-normalized gene expression data employed in the eQTL study, to evaluate the association between previously identified eGenes (n = 805) and corresponding eQTLs (n = 4558) with overall survival (OS). The tested alternative alleles and corresponding genotypes in survival analysis were the same as the alternative alleles used in eQTL analysis. Clinical parameters, including gender, age, and cancer stage at diagnosis (Tumor size, Node involvement, Metastasis status), were also assessed for their potential prognostic effects in both cohorts.

Significant survival-associated clinical factors identified from the univariate analysis were included as adjusted covariates in multivariate Cox models to determine the independent prognostic value of each eGene and eQTL.

For both univariate and multivariate Cox models, the eQTLs were employed by two inheritance models to analyze survival associations allele-specifically:

1) Allelic Model: Assumes a linear effect of the number of minor alleles on the hazard ratio, treating SNP genotypes as numeric values corresponding to the number of minor alleles. Only one Hazard ratio (HR) is given in this model.

2) Codominant Model: Treats SNP genotypes as categorical variables, where heterozygous and homozygous states are each compared to a reference category. The HR is calculated separately for heterozygous (HRhet) and homozygous (HRhom).

All analysis was performed by the R package 'survival' with statistical significance set as $P < 0.05$ (Wald-test). eQTLs with fewer than three individuals in both the heterozygous and homozygous genotype categories were excluded. Multiple testing corrections were performed using Benjamin-Hochberg (BH) to get solid signals with FDR < 0.05. Negative coefficients (coef <0, HR < 1) to death were interpreted as favourable effects, indicating that higher expression values for genes or the presence of alternative alleles were associated with a better prognosis. Conversely, positive coefficients (coef > 0, HR > 1) were considered unfavourable effects, indicating that higher gene expression or the presence of alternative alleles was associated with a worse prognosis.

To further explore the consistency of prognostic values of eGene-eQTL pairs, we applied a parallel survival analysis integrating the slope of allelic effect deprived of eQTLs association analysis. We hypothesise that the linear regression slope of an eQTL and its corresponding eGene should have similar effects regarding the survival outcome. Specifically, for an eQTL with a positive slope with favorable prognostic genes, we suppose this eQTL should have consistent effects and be favorable for patients' survival, and for an eQTL with a negative slope, the eGene should have converse effects for patients' survival. The eQTLs that have consensus prognostic attributes are then regarded as promising SNP/Indels that regulate the expression of the prognostic eGene. The eGenes-eQTLs pairs with the same allelic effect directions on the JP and TCGA cohorts might suggest stronger confidence in the associations. The eQTLs with a P-value < 0.05 in any Cox models described above and with a consistent prognosis on corresponding eGenes in the JP cohort were selected for validation in the TCGA cohort. The TCGA cohort lacks Node involvement and metastasis status, so only Tumor size was included in the adjusted Cox analysis.

To comprehensively evaluate the replicability of survival associations across populations, we implemented a two-tiered replication framework: strict and lenient replication criteria. For strict replication, we required both statistical significance (P < 0.05) and consistent effect direction between the JP cohort and TCGA data. For lenient replication, we focused solely on directional consistency of effects, regardless of statistical significance in TCGA.

In addition, the Kaplan-Meier (KM) analysis with a log-rank test was applied to visualize the prognosis effects of eGenes and eSNPs. Each given eGene was stratified into high and low-expression groups based on TPM values across all samples. The optimal cut-off value was determined by testing TPM values ranging from the 20th to 80th percentiles, with the cut-off yielding the lowest log-rank P-value selected. Variant-level KM analysis was performed by stratifying eQTLs by genotypes that classifies as 3 categorical values: 0 represents reference, 1 represents heterozygous variants and 1 represents homozygous variants. The log-rank p-value was calculated based on the KM analysis.

## 3. Results

### 3.1. Cis-eQTL analysis of the JP cohort

To understand how genetic variation influences gene expression in Japanese ccRCC patients and potentially identify population-specific regulatory variants, we performed a comprehensive cis-eQTL analysis. As data quality is crucial for reliable eQTL mapping, we first implemented rigorous quality control steps following GATK and GTEx pipeline recommendations to filter out low-quality variants and unreliable genotypes.

S1 Table shows the number of remaining variants of the JP cohort in each step of quality control. A total of 284774 variants were kept after variants QC filtering, including 244803 SNPs, 17346 insertions, and 22625 deletions. Of these, 283,376 variants (~99.5%) fell within the ± 1 Mb windows around gene TSSs and were included in the FastQTL analysis. The majority of them being benign variants based on SnpEff molecular consequences annotation, while only approximately 0.18% of variants were annotated as 'HIGH' impact by SnpEff, representing the predicted deleterious effects on the gene. Inside ±1 Mb windows, 60.9% of tested variants fall in intronic region, 16.3% of them in coding region and 8.2% in regulatory region (S2A Fig). S2B Fig displays the distribution of variant distances from TSSs. The count distribution of tested variants per gene is shown in S2C Fig. A chromosome-level density plot depicting the overall genomic distribution of tested variants is also provided as S2D Fig.

To perform cis-eQTL analyses on the JP cohort (n = 100), the linear regression models were applied for all testing gene-genotype pairs by FastQTL. A total of 25,508 genes with 8495717 pairs of gene-SNP/Indels were tested for the associations across all variants inside the cis window of 1 Mb(S3 Fig), which excluded 180 genes that lacked variant information The significantly associated genes with the corresponding top-eQTL were shown in the Manhattan plot ranking by q-value (Fig 2A). We identified 805 eGenes with FDR values < 0.05 from FastQTL results (S2 Table) which are labelled by the red dot in Fig 2A. The top ten eGenes ranked by FDR value were annotated with the gene names in the Manhattan plot and their most significant associated eQTLs (top-eQTLs) were listed in Table 1. The top-4 most significant eGenes, LINC01291, ERAP2, XRRA1 and RPS26 with their top-eQTLs were shown in Fig 2B, where further linear regression analysis was performed using addictive, dominant and recessive models to confirm their significant differential expressions among their associated eQTL signals. Similarly, among our top-100 findings ranked by q-value, all of them passed this sensitivity analysis except for FAM171B. Of the 805 eGenes tested, 46 failed to pass the sensitivity analysis; these genes are highlighted and listed in S2 Table and were excluded from downstream cross-cohort survival analysis on validation phase. Among the top 4 eGenes, some of them were reported as regulatory factors in cancers, such as LINC1291, which is an important factor that promotes the aggressive properties of melanoma [36]. Notably, only 56% of the eGenes are protein-coding genes, and the rest are long non-coding RNA (lncRNA) genes (23%), pseudogenes (12%) and other gene subtypes (10%) (Fig 3A).

In total, 5228 significant eQTL-eGene pairs including 4558 unique eQTL loci were identified (FDR < 0.05) within the cis-window of 1 Mb (S3 Table). Among the 4558 unique eQTLs, 90% of them only target to one eGene and 10% of eQTLs target more than one eGenes (S4 Fig). There were 4045 SNPs, 201 insertions and 321 deletions classified by variant types. Based on the VEP annotations of most severe consequences, the unique eQTLs were mostly intronic (51%) followed by exon variants for non-coding transcripts (18%) (Fig 3B). Fig 3C illustrates the negative association between

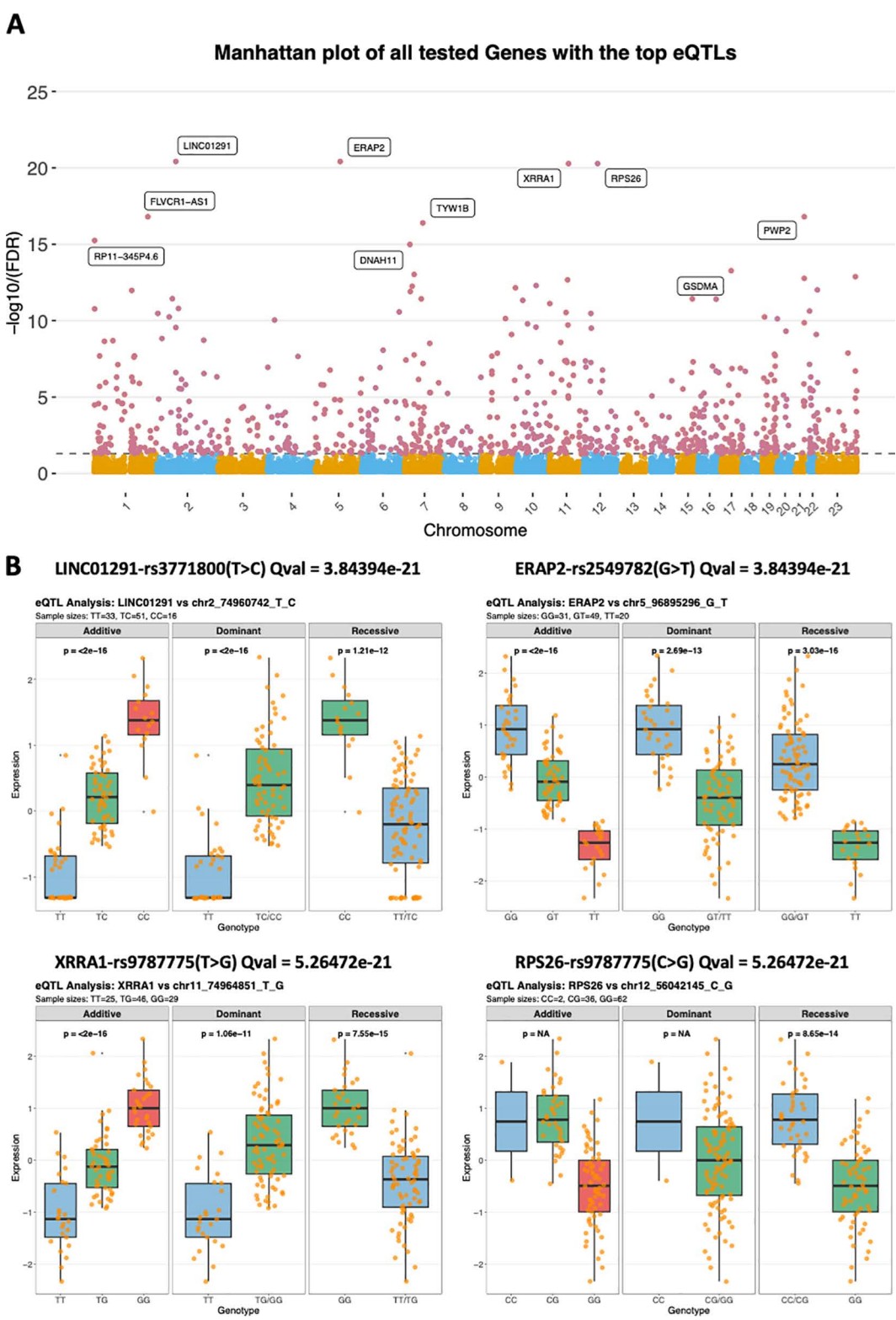

**Fig 2. eQTL analysis on the JP ccRCC cohort and the most significant results. (A)** Manhattan plot of all tested genes with their top eQTLs, ranked by q-value. Red dots represent the 805 significant eGenes (FDR < 0.05), with the top ten eGenes, ranked by FDR value, annotated by gene names on

the plot. This plot highlights the most significant genetic associations influencing gene expression in the cohort. **(B)** Box plots showing expression levels of the top four eGenes—LINC01291, ERAP2, XRRA1, and RPS26—based on their top-eQTL genotypes that tested by addictive, dominant and recessive genetic models. Each plot illustrates the differential expression across genotypes, with q-values indicating the strength of association from FastQTL results and pvalue generated from simple linear regression testing for three genetic models (Any group of genotypes lower than three were labeled as NA). Some genes, such as LINC01291, have known roles in cancer regulation, with prior studies linking them to aggressive cancer traits.

**Table 1. The ten most significant eGenes identified in the ccRCC cohort.**

| gene name | gene chr | variant id | rs id | TSS | MAF | Qval(FDR) |
|---|---|---|---|---|---|---|
| LINC01291 | chr2 | chr2_74960742_T_C | rs3771800 | 42594 | 0.4462370 | 3.84394e-21 |
| ERAP2 | chr5 | chr5_96895296_G_T | rs2549782 | 19357 | 0.4494950 | 3.84394e-21 |
| XRRA1 | chr11 | chr11_74964851_T_G | rs9787775 | 15651 | 0.4468090 | 5.26472e-21 |
| RPS26 | chr12 | chr12_56042145_C_G | rs1131017 | 292 | 0.1919190 | 5.26472e-21 |
| FLVCR1-AS1 | chr1 | chr1_212859226_T_A | rs12023052 | 1138 | 0.4646460 | 1.58394e-17 |
| PWP2 | chr21 | chr21_44093927_T_C | rs8132918 | -13363 | 0.3826530 | 1.58394e-17 |
| TYW1B | chr7 | chr7_72768844_C_T | None | -59354 | 0.2070710 | 4.02365e-17 |
| RP11-345P4.6 | chr1 | chr1_1676955_C_T | rs146877222 | 4965 | 0.4578950 | 5.68232e-16 |
| DNAH11 | chr7 | chr7_21545202_A_G | rs7781669 | 1987 | 0.3092780 | 1.03004e-15 |
| GSDMA | chr17 | chr17_39972461_G_A | rs3859191 | 19198 | 0.4020620 | 5.40745e-14 |

effect allele frequency and absolute eQTL effect size (beta-coefficients). Spearman correlation analysis revealed a significant negative correlation for both all significant eQTL pairs ($\rho = -0.451$, $p < 1.5e-260$) and top eQTL per gene ($\rho = -0.599$, $p < 2.1e-79$). Most of the eQTLs were located close to the transcription start site (TSS) of associated eGenes, especially for the top-eQTLs (Fig 3D).

### 3.2. Comparison of eQTL analysis across existing databases

To investigate the compositions of the common eGenes and eQTLs identified in the JP cohort, we incorporated two public cis-eQTLs datasets, GTEx Kidney cortex deprived of healthy kidney tissue and Pan-cancer database deprived of ccRCC samples. A total of 1260 eGenes, 111169 significant pairs and 76365 unique eQTLs (GTEx-eQTLs) were obtained from GTEx V8. A total of 8739 eGenes, 521072 significant pairs and 410720 unique eQTLs (Pan-cancer eQTLs) were obtained from the Pan-cancer database.

The overlap of identified eGenes among the 3 datasets is shown in Fig 3E. On the eGenes level, 287 eGenes overlapped between the JP cohort and the GTEx (35.7%, $p = 5.235324e-173$, hypergeometric test). Among the 805 eGenes in the JP cohort, 321 eGenes are shared in both the JP cohort and the Pan-cancer study (39.9%, $p = 4.19037e-4$, hypergeometric test).

We also compared the eQTLs identified in the JP cohort with the two databases. We found that there are 4015 (among 76365) GTEx eQTLs and 22920 (among 410720) Pan-cancer eQTLs that overlapped with the JP variants set, respectively. A total of 1452 eQTLs appeared both in the JP cohort and GTEx which accounts for 31.86% of the total number of JP cancer-eQTLs (S5A Fig). Compared with the Pan-cancer study, there are 2265 (49.7%) of them also identified in Pan-cancer ccRCC eQTLs, which showed the similarities between the two cohorts with different ethnic groups (S5B Fig). There are also more than 50% of eQTL identified in the JP cohort that is not reported in the Pan-cancer study that being the unique finding for this study. The results suggest that the cancer eQTLs we identified in the JP cohort significantly overlapped with the eQTLs identified by other studies while our results explored a proportion of unique eQTLs either by cancer-specific expression or ethnic differences.

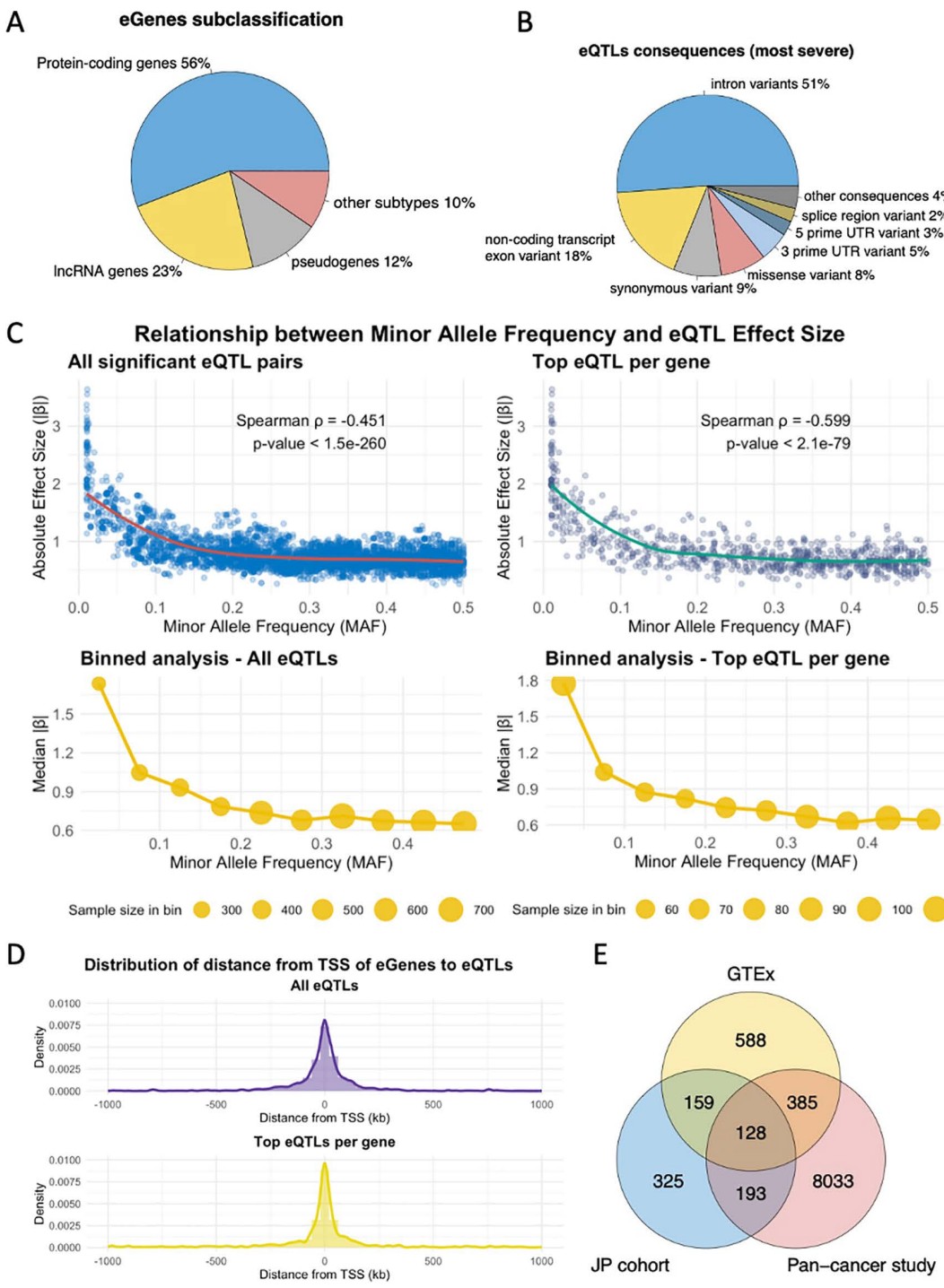

**Fig 3. Characterization of eGenes and eQTLs in the JP Cohort and Comparison with Public Datasets.** (A) Subclassification of eGenes in the JP cohort. The majority of eGenes are protein-coding (56%), with a significant proportion classified as long non-coding RNA (lncRNA) genes (23%), pseudogenes (12%), and other subtypes (10%). (B) Distribution of the most severe consequences of eQTLs as annotated by VEP. The majority of eQTLs are intronic variants (51%), followed by non-coding transcript exon variants (18%), synonymous variants (9%), missense variants (8%), and other minor categories. (C) Relationship between Minor Allele Frequency (MAF) and absolute eQTL effect size. Scatter plots display the relationship between MAF and absolute β coefficients for all significant eQTLs (left) and the top eQTL per gene (right). Each point represents a single eQTL signal, with LOESS smoothing curves (red and green) depicting the overall trend. Binned box plots show the distribution of absolute effect sizes across MAF ranges

(0-0.05, 0.05-0.1, 0.1-0.15, etc.). Spearman correlation coefficients indicate significant negative correlations. The notably high effect sizes observed at MAF ≈ 0.01 correspond to variants with low minor allele count (n = 2). (D) Density plot showing the distance distribution of eQTLs from the transcription start site (TSS) of associated eGenes. Top-eQTLs are predominantly located near the TSS, indicating close cis-regulatory relationships. (E) Venn diagram comparing eGenes identified in the JP cohort with those from GTEx Kidney cortex and Pan-cancer eQTL datasets. A total of 287 eGenes are shared between the JP cohort and GTEx, and 321 eGenes overlap between the JP cohort and Pan-cancer dataset, indicating a substantial commonality and uniqueness in eGene profiles across datasets.

### 3.3. Somatic mutations and recurrent somatically mutated cohort-specific eGenes

To understand the relationship between somatic mutations and eQTL regulation in Japanese ccRCC patients, and to identify potential functional connections between germline variants and somatic alterations, we performed comprehensive somatic mutation analysis. We focused particularly on identifying recurrently mutated genes that overlapped with our eQTL findings, as these could represent important regulatory nodes in ccRCC development.

Only mutations that remained after running FilterMutectCalls were used for statistics and downstream analysis. A total of 23997 somatic variants remained, including 21046 SNPs,352 Multiplenucleotide polymorphisms (MNPs),610 insertions, and 1989 deletions. Variants with 'HIGH' impact represent 2.32% of total somatic mutations. Missense mutations accounted for a greater proportion of the mutations (71.281%) than silent mutations (24.753%). Most patients have around 250–400 somatic mutation loads which are logtransformed to 2.42.6 (S6 Fig). A total of 712 mutated genes with at least five recurrent mutations have been identified (S4 Table). Gene-level somatic mutation frequencies are presented in Fig 4A, with intergenic regions and ambiguous multi-gene annotations excluded. BAGE2, a long noncoding RNA (lncRNA) gene often related to melanoma, had the highest frequency of somatic mutations (67 mutations). The second most frequent mutated gene was VHL (54 mutations) which is the most convincing risk gene for ccRCC predispositions. The third most frequently mutated gene was TTN (51 mutations), which encodes the protein Titin. PBRM1 ranked fourth with 44 mutations. Other frequently mutated genes included SETD2 (15 mutations) and BAP1 (12 mutations), both of which have been extensively reported in ccRCC studies and cases. In addition to these, several other genes from the JP cohort were among the top 10 mutated genes based on The Cancer Genome Atlas (TCGA) ccRCC data. These included MUC16 (33 mutations), MTOR (16 mutations), DNAH9 (12 mutations), HMCN1 (15 mutations), and KDM5C (7 mutations).

A total of 19 significant eGenes of the JP cohort overlapped with highly recurrent somatic mutated genes (at least 5 times), as shown in Fig 4B and 4C. Among them, DNAH11 showed the highest somatic mutation load (19 mutations) and the most significant eQTL signals (by q-value).

### 3.4. Survival analysis to identify prognostic eGenes and eQTLs in the JP ccRCC cohort

To evaluate the clinical relevance of identified eQTLs and their associated genes, we conducted a comprehensive survival analysis. Our approach aimed to identify genetic variants and gene expression patterns that could predict patient outcomes, with careful consideration of both clinical factors and different genetic inheritance models.

Firstly, we investigated the gender, age, tumor size (T), the extent of spread to the lymph nodes (N), and the presence of metastasis (M) by univariate Cox analysis on the JP cohort. The results (S7 Fig) indicated that age and gender are not significantly associated with OS, while Tumor stage, spread to lymph nodes and metastasis are significantly associated with OS (P < 0.001) in the JP cohort.

To further estimate the survival associations, we applied both univariate Cox regression model (Cox) and multivariate Cox models that were adjusted by TNM stages to the 805 eGenes and 4558 eQTLs identified within the JP cohort. In the context of eGenes, after multiple testing correction by the BH procedure, there are 7 significant prognostic eGenes identified in the univariate Cox analysis and 3 prognostic eGenes in the adjusted Cox analysis (Table 2), which sum to a total of 9 unique eGenes. In the context of eSNPs, there were no significant prognostic signals on the threshold of FDR < 0.05.

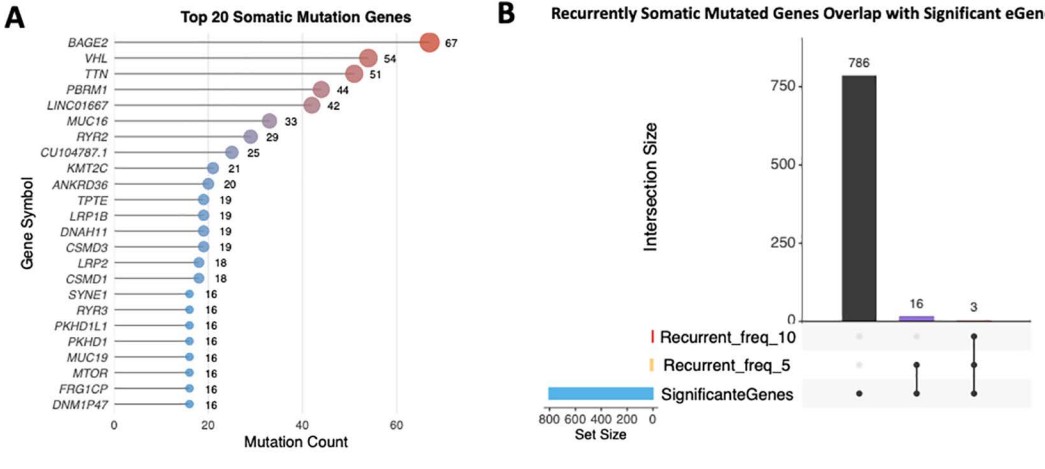

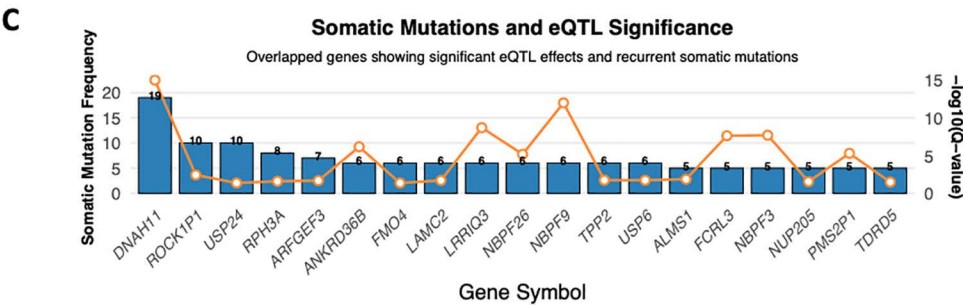

**Fig 4. Characterization of Somatic Recurrent Mutated Genes with eQTL associations. (A)** Top 20 somatically mutated genes ranked by mutation frequency across the JP cohort. The most recurrently mutated genes include VHL (67 mutations), BAGE2 (54 mutations), and TTN (51 mutations) **B)** Upset plot visualizing the overlap between significant eGenes (805 genes) and recurrently mutated genes at two frequency thresholds (≥5 and ≥10). A total of 19 somatically recurrent mutated genes overlapped with eGens signals. **(C)** Dual-axis visualization of somatic mutation frequencies and eQTL significance for 19 genes that overlap. Blue bars represent somatic mutation counts (left y-axis), while orange points with connecting lines indicate -log10(Q-value) for eQTL significance (right y-axis). Notable genes include DNAH11 (19 mutations) demonstrating varying levels of somatic burden alongside their roles as significant eGenes.

**Table 2. The significant prognostic eGenes (FDR<0.05) from Cox analysis of JP cohort (Unadjusted & Adjusted).**

| gene_name | COX_coef | P | FDR | mean_TPM | simple_gene_id | gene_biotype | Models |
|---|---|---|---|---|---|---|---|
| HEATR3 | 0.118081510803151 | 0.000101734787438635 | 0.0117521989575024 | 14.5714431 | ENSG00000155393 | protein_coding | Unadjusted Cox (Prognosis~eGenes) |
| TUBA1C | 0.0136919860126823 | 0.000102193034413064 | 0.0117521989575024 | 48.716941 | ENSG00000167553 | protein_coding | |
| RP11-218F10.3 | 0.516706512257168 | 0.000133331833116278 | 0.0134165157073255 | 1.09183491 | ENSG00000273449 | lncRNA | |
| CPNE1 | 0.0124393189847147 | 0.000295297023621351 | 0.0264126782239097 | 89.079656 | ENSG00000214078 | protein_coding | |
| MMP16 | 0.145675293952334 | 0.000333449662722653 | 0.0268426978491736 | 1.18333664 | ENSG00000156103 | protein_coding | |
| RAET1E | -3.47214258299286 | 0.000451638384273497 | 0.0330517181218332 | 0.50258754 | ENSG00000164520 | protein_coding | |
| GEN1 | 0.306880495210678 | 0.000583502480514238 | 0.0391432914011635 | 3.09747835 | ENSG00000178295 | protein_coding | |
| CENPU | 0.0873760607542322 | 0.000300868001600201 | 0.0453022028183162 | 8.39568621 | ENSG00000151725 | protein_coding | Adjusted Cox (Prognosis~eGenes +sex+age+TNM) |
| STMP1 | 0.0843195495340997 | 0.000186453679484458 | 0.0375238029962471 | 27.600688 | ENSG00000243317 | protein_coding | |
| TUBA1C | 0.0154766850518572 | 0.000337656170074406 | 0.0453022028183162 | 48.716941 | ENSG00000167553 | protein_coding | |

We further performed a novel approach of exploratory analysis of eGene-eSNPs pairs with their potential prognostic effects, especially on the hypothesis that important eGene-eSNPs pairs would demonstrate consistent prognosis patterns at both gene and SNP levels. Furthermore, we aimed to find those consistent pairs across two independent cohorts. The hypothesis and results were summarized in Fig 5A and 5B. In this approach, we kept signals with a p-value <0.05 as explorable candidates. The univariate Cox analysis demonstrated statistical significance for 158 eGenes, while the adjusted Cox regression model identified 82 significant eGenes, of which 53 were significant in both unadjusted and adjusted models, and a union set of 187 eGenes was defined as potential prognostic eGenes (S5 Table). The unadjusted and adjusted models give the same prognostic effect on these genes. Among these, there were 105 favourable eGenes and 82 unfavourable eGenes.

The eQTL analysis was performed using both the codominant and allelic models, under unadjusted and adjusted conditions. Any eQTLs with both heterozygous and homozygous genotypes observed in fewer than three individuals were excluded. Firstly, the hazard ratios (HR) and 95% confidence intervals (CI) for all of the top-eQTLs were reported in S6 Table. Values for some eQTLs with extreme HRs were set to NA due to limited sample size, where all carriers of certain genotypes had the same outcome (all deceased/alive). Notably, rs3771800(T>C)-LINC01291, as one of the top-4 eQTLs (Fig 2), had a protective signal (HR<1) with a p-value <0.05 under adjusted models.

Then, we focused on the eQTLs selected from either model based on the threshold of $P < 0.05$. A total of 367 and 531 eQTLs were found to be significant in unadjusted and adjusted Cox models, respectively, resulting in 711 significant eQTLs overall (S7 Table). Of these, 526 showed consistent prognostic coefficient directions across both models, while 185 showed differing directions. The full list of prognostic eQTLs, ranked by P-value along with their coefficients, is provided in S7 Table. Kaplan-Meier (KM) plots were generated to visualize the prognostic effects of all eGenes and eQTLs. As an example, the four most significant survival-associated eGenes and eQTLs, ranked by log-rank P-value, are shown in S8 Fig.

To further explore the consistency of prognostic values of eGene-eQTL pairs, we performed a parallel survival analysis integrating the slope of allelic effect derived from eQTLs association analysis. We hypothesized that the linear regression slope of an eQTL and its corresponding eGene should have similar effects regarding the survival outcome. Specifically, a positive slope for favorable prognostic genes suggests that the eQTL would positively impact survival, while a negative slope implies opposite effects between the pairs (Fig 5B). eQTLs showing consistent prognostic values were considered promising SNP/Indels regulating prognostic eGenes. As a result, we found 54 eGenes (Fig 5C) with 223 eQTLs within 329 pairs (S8 Table) that are significantly associated with the OS on both the eGene and the eQTL levels. Of the 54 genes, TDRD5 was the only gene also found as a recurrent somatically mutated gene, with five mutations. These target pairs were specific to the JP cohort and were validated using the TCGA cohort on the downstream analysis.

### 3.5. Validation phase of survival analysis on TCGA cohort

To validate the potential survival associations of the eQTLs identified in the JP cohort, we performed the same survival analysis on 223 eQTLs using whole-exome sequencing (WES) and clinical data from 287 TCGA patients. Of these, 209 variants were retrieved based on genomic location and allele information.

To further assess replication across cohorts, we applied a lenient criterion that examined whether the variants significantly associated with survival in the JP cohort showed consistent effect directions in the TCGA cohort, regardless of statistical significance. From this approach, a total of 181 eQTLs with concordant effect directions between cohorts were highlighted as orange dots in Fig 6A.

We then focused on the strict replicated results based on JP pairwise passed signals. The initial survival analysis revealed 15 significant and 24 significant variants (P < 0.05) from unadjusted and adjusted Cox models, respectively. After filtering SNPs with at least three genotype counts in both cohorts and comparing the allelic effects, we identified 19 eGene-eQTL pairs with consistent allelic directions and prognosis effects across models, which are detailed in S9 Table.

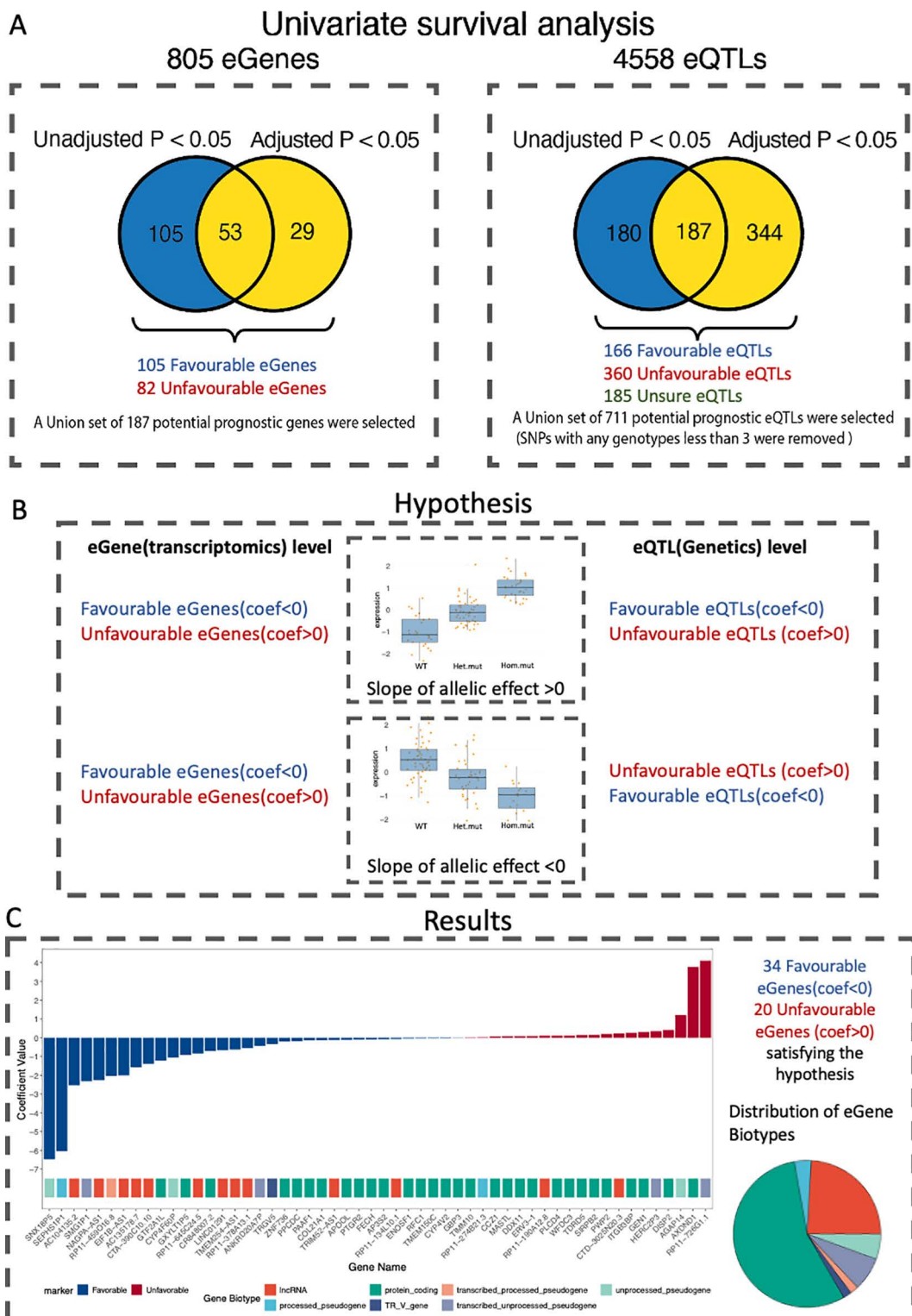

**Fig 5. Survival Analysis of eGenes and eQTLs in the JP Cohort.** (A) Summary of univariate and adjusted Cox regression models applied to 805 eGenes and 4558 eQTLs. Univariate analysis (by p-value <0.05 cutoff) identified 158 significant eGenes, with 82 remaining significant in the adjusted model (TNM stage adjusted). A union set of 187 eGenes was defined as prognostic, with 105 favorable and 82 unfavorable based on their survival

association. For eQTLs, 711 significant variants were identified, with 166 favorable and 360 unfavorable, forming a prognostic set after filtering. (B) Hypothesis testing of eGene-eQTL pairs for prognostic consistency. We hypothesized that favorable and unfavorable prognostic eGenes should have eQTLs with consistent allelic effect slopes on survival. Positive slopes indicate favorable effects, while negative slopes suggest adverse effects, allowing identification of eQTLs with consistent prognostic values. (C) Results of hypothesis testing, showing 54 prognostic eGenes with 223 associated eQTLs forming 329 consistent eGene-eQTL pairs. A bar plot illustrates the distribution of favorable and unfavorable eGenes based on coefficient direction, with a biotype pie chart indicating the composition of identified prognostic eGenes.

These included eight strict replicated eSNPs linked to 11 eGenes (Table 3, Fig 6B for JP, Fig 6C for TCGA). Notably, four eGenes: DISP2 (mutated twice), ERV3–1, ANKRD20A7P, and CYP4F60P (each mutated once)—were among the 11 eGenes found to be also somatically mutated. Of these somatic mutations, ERV3–1 (c.592A>T, p.Asn198Tyr) is the only variant located in a protein-coding region.

While some SNPs are located extremely close, after calculating their LD, we identified a total of 5 unique genomic loci (LD $r^2 < 0.1$) to be strongly associated with survival on an ethnically wide range. We searched the eQTLs on RegulomeDB to evaluate the regulatory effects of these SNPs (Table 3). There is no GWAS co-localization signal associated with ccRCC as searched by LDtrait Tool. However, potential regulatory effects were predicted through RegulomeDB. We found that the rs3803360 (DISP2) and rs7806567 (SEPHS1P1) have the highest level of 1b among them which indicates the inquired SNP is with the annotation of eQTL/caQTL, TF binding, any motif, Footprint and chromatin accessibility peak. The query of the Chromatin stage shows that the region of rs3803360 falls in the results of Quiescent/Low the most, while the Weak Repressed PolyComb has the second most frequent results in kidney tissues. For rs7806567, weak enhancer gets the most results and flanking TSS is second. For other SNPs, the regulatory effects are predicted as 1f which is also relatively high. The variants' annotation indicated that rs34639489 (ERV3–1) is a missense variant that might cause deleterious effects to the corresponding protein (SIFT = 0, Polyphen = 0.957).

Fig 7 illustrates the pairwise prognostic effects of validated SNP-Gene pairs across both cohorts. As an example, for rs28612439, the AA genotype is associated with lower expression of ANKRD20A7P in the JP cohort and GTEx database in kidney tissues with the same allelic effect directions (Figs 7A and S9A), while the GG genotype shows higher expression. ANKRD20A7P is a favorable prognostic gene (KM P = 0.0003, Cox P = 0.003/0.092 for unadjusted/adjusted), with lower expression linked to a worse prognosis. Genotype-level analysis shows the AA genotype correlates with poorer prognosis in both the JP (KM P = 0.019, Cox P = 0.032/0.025) and TCGA cohorts (KM P = 0.038, Cox P = 0.024/0.006) (Figs 6 and 7A). This suggests rs28612439-ANKRD20A7P as a potential regulatory and risk factor in ccRCC progression. Similarly, rs34639489 is also an eQTL that is significantly associated with ERV3–1 for the JP cohort and GTEx database, with the same allelic effect directions (Figs 7B and S9B). The rs34639489 TT genotype is associated with a lower expression level of ERV3–1. ERV3–1 is identified as an unfavorable prognostic gene (KM P = 0.0154, Cox P = 0.002/0.019), and higher expression is associated with worse survival. Since the slope is negative, rs34639489 is associated with better survival for both the JP cohort (KM P = 0.103, Cox P = 0.044/0.026) and the TCGA cohort (KM P = 0.092, Cox P = 0.087/0.045) (Figs 6 and 7B).

### 3.6. Comparison with Published TWAS Highlights Shared ccRCC-relevant Genes.

To further validate our findings, we compared our results with the major gene lists from a transcriptome- and proteome-wide association study for renal cell carcinoma (RCC) [37]. This comparison identified 15 genes overlapping with JP ccRCC eGenes, including 4 that were also among our candidate prognostic eGenes (S10A–C Fig). Specifically, we found that RCCD1, MLF2, HEATR3, and EIF1B-AS1 as both survival-associated eGenes (p < 0.05) and TWAS findings that associated with RCC and ccRCC disease risk. While the overall overlap is limited, it highlights a set of genes that offer additional candidates for future investigation in the molecular mechanisms of ccRCC.

                                    

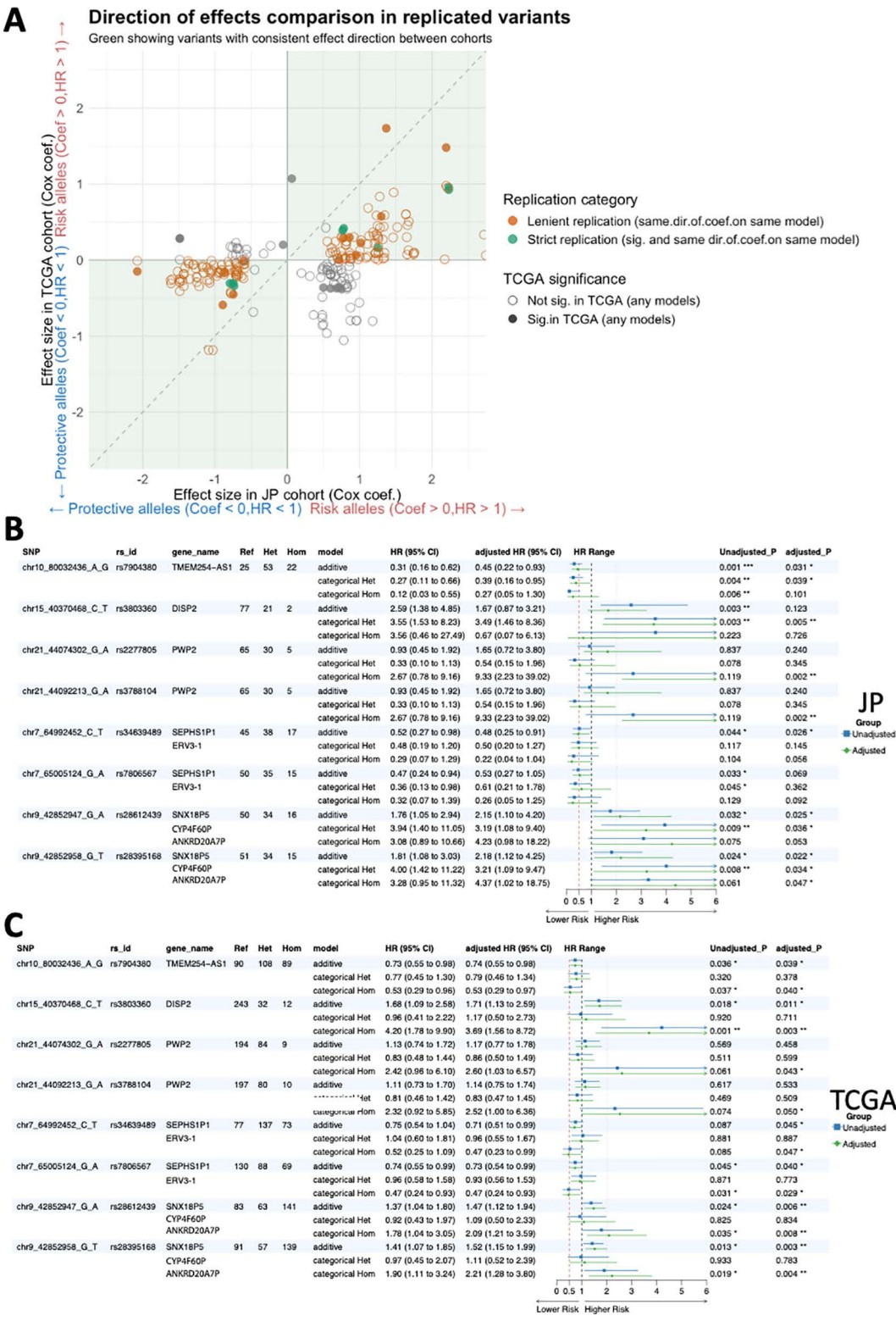

**Fig 6. Validation of Survival-Associated eQTLs in JP and TCGA Cohorts.** (A) Cross-population validation of variant effects on survival outcomes. Scatter plot comparing the effect sizes (β coefficients) of survival-associated variants between JP ccRCC cohort (JP) and TCGA ccRCC cohort. Each point represents a single variant-expression association, with the x-axis showing effect sizes in the JP cohort and y-axis showing effects in the TCGA

cohort. The diagonal lines indicate perfect directional concordance. Green points represent variants with eight strict replication (significant in both cohorts with consistent direction), orange points represent lenient replication (consistent direction regardless of statistical significance in TCGA), and grey points indicate discordant directional effects. (B-C) Eight eSNPs linked to 11 eGenes demonstrated consistent survival associations and allelic direction between the JP (B) and the TCGA (C) cohort (by strict replication). The consistent allelic direction and HR across models suggest robust survival associations for these variants.

**Table 3. The final pairs of eQTL-eGenes with pairwise prognosis effect between JP and TCGA kidney cancer cohort.**

| RS_ID | Genotype | Variant Consequences | Regu-lome | Gene Name | Gene Biotype | Gene Prognosis | JP-QTLs Slope | Satisfied Cox Models on JP&TCGA |
|---|---|---|---|---|---|---|---|---|
| rs7904380 | chr10_80032436_A_G | Intergenic | 1f | TMEM254-AS1 | lncRNA | Favorable | 0.9396 | Additive, Cate. Hom, Additive Adj. |
| rs3803360 | chr15_40370468_C_T | 3'UTR | 1b | DISP2 | Protein coding | Unfavorable | 1.04491 | Additive |
| rs2277805 | chr21_44074302_G_A | Intron | 1f | PWP2 | Protein coding | Unfavorable | 1.10637 | Cate. Hom Adj. |
| rs3788104 | chr21_44092213_G_A | Intron | 1f | PWP2 | Protein coding | Unfavorable | 1.09492 | Cate. Hom Adj. |
| rs34639489 | chr7_64992452_C_T | Missense | 1f | SEPHS1P1 | Pseudogene | Favorable | 0.772952 | Additive Adj. |
| | | | | ERV3–1 | Protein coding | Unfavorable | -0.470105 | Additive Adj. |
| rs7806567 | chr7_65005124_G_A | Non-coding | 1b | SEPHS1P1 | Pseudogene | Favorable | 0.712683 | Additive |
| rs28612439 | chr9_42852947_G_A | Intron | 1f | CR848007.2 | Pseudogene | Favorable | -0.812756 | Additive, Additive Adj. |
| | | | | CYP4F60P | Pseudogene | Favorable | -0.761645 | Additive, Additive Adj. |
| | | | | ANKRD20A7P | Pseudogene | Favorable | -0.854169 | Additive, Additive Adj. |
| | | | | SNX18P5 | Pseudogene | Favorable | -0.563103 | Additive, Additive Adj. |
| | | | | RP11-459O16.8 | Pseudogene | Favorable | -0.728779 | Additive, Additive Adj. |
| | | | | GXYLT1P5 | Pseudogene | Favorable | -0.820142 | Additive, Additive Adj. |
| rs28395168 | chr9_42852958_G_T | Intron | 1f | CR848007 | Pseudogene | Favorable | -0.815241 | Additive, Additive Adj., Cate. Hom Adj. |
| | | | | CYP4F60P | Pseudogene | Favorable | -0.769136 | Additive, Additive Adj., Cate. Hom Adj. |
| | | | | ANKRD20A7P | Pseudogene | Favorable | -0.855723 | Additive, Additive Adj., Cate. Hom Adj. |
| | | | | SNX18P5 | Pseudogene | Favorable | -0.539548 | Additive, Additive Adj., Cate. Hom Adj. |
| | | | | RP11-459O16.8 | Pseudogene | Favorable | -0.743183 | Additive, Additive Adj., Cate. Hom Adj. |
| | | | | GXYLT1P5 | Pseudogene | Favorable | -0.820471 | Additive, Additive Adj., Cate. Hom Adj. |

## 4. Discussion

Systems biology provides the methodology of intergrating multi-omics data to gain deeper insights into precision medicine [38]. The giant numbers of GWAS and QTL mapping studies are aiming at building clear networks of genetics-transcriptomics-phenomics. Previous eQTL studies have predominantly focused on blood-derived cells, cell lines [39] and normal human tissues as represented by the GTEx project [13]. However, studies have shown that eQTLs identified in cancer tissues are often cancer-specific compared to those from normal tissues, as demonstrated in colorectal cancer [40] and breast cancer [41]. Despite these advancements, cancer-specific eQTL data remain limited, particularly for

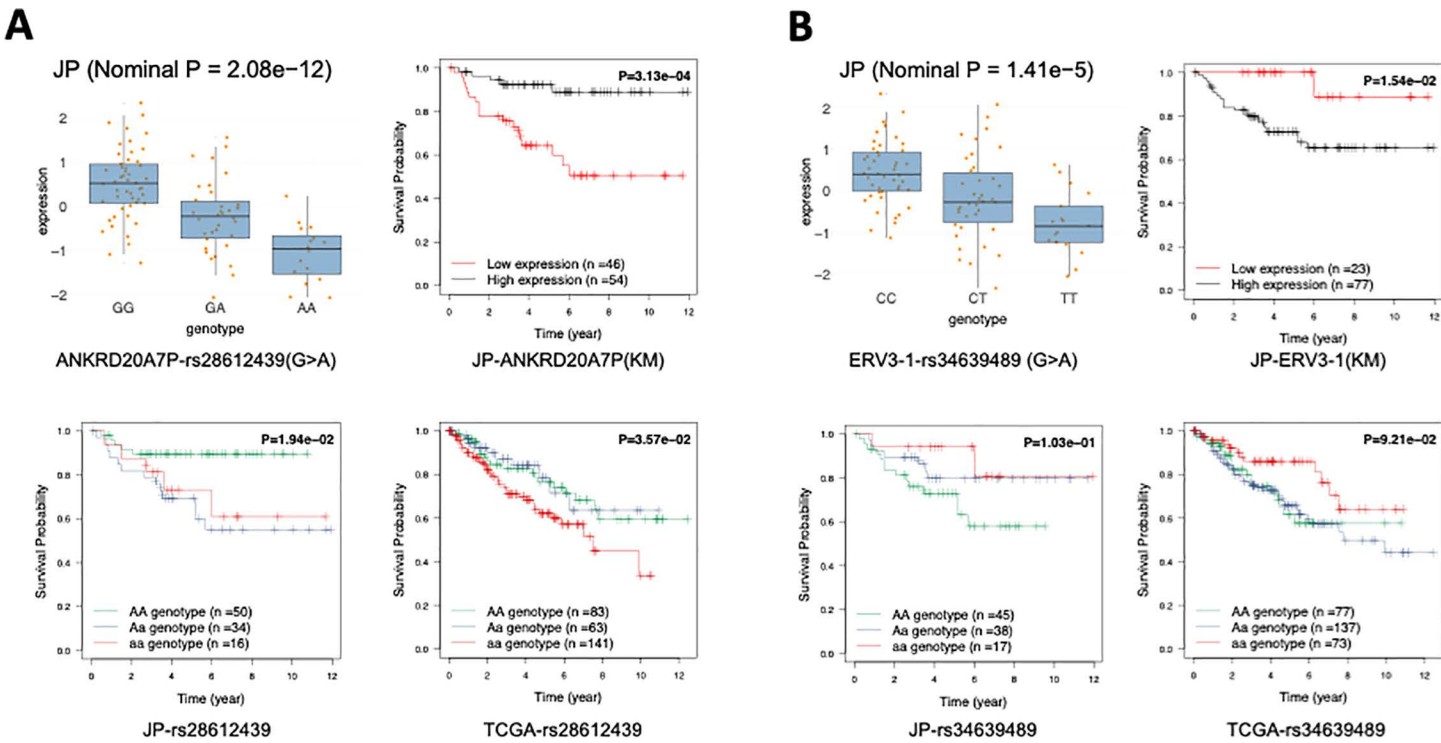

**Fig 7. Pairwise Prognostic Effects of Validated SNP-Gene Pairs in JP and TCGA Cohorts.** (A) Association of rs28612439 with ANKRD20A7P expression and prognosis. In both the JP cohort and GTEx kidney cortex data (S9A Fig), the AA genotype is associated with lower ANKRD20A7P expression, while the GG genotype correlates with higher expression. ANKRD20A7P is a favorable prognostic gene, with lower expression linked to worse outcomes (KM P=0.0003, Cox P=0.003/0.092 for unadjusted/adjusted). Genotype-level analysis shows that the AA genotype is associated with poorer prognosis in both the JP (KM P=0.019, Cox P=0.032/0.025 for unadjusted/adjusted) and TCGA cohorts (KM P=0.038, Cox P=0.024/0.006), suggesting rs28612439-ANKRD20A7P as a potential regulatory risk factor in ccRCC progression. (B) Association of rs34639489 with ERV3-1 expression and prognosis. The TT genotype is associated with lower ERV3-1 expression in both the JP cohort and GTEx kidney cortex data (S9B Fig). ERV3-1 is an unfavorable prognostic gene, where higher expression is linked to worse survival outcomes (KM P=0.0154, Cox P=0.002/0.019). The negative slope indicates that the variant is associated with better survival, as observed in both the JP cohort (KM P=0.103, Cox P=0.044/0.026) and the TCGA cohort (KM P=0.092, Cox P=0.087/0.045). This suggests that rs34639489-ERV3-1 may play a protective role in ccRCC.

non-European populations, as most eQTL studies have predominantly included individuals of European ancestry, similar to other genetic studies. Furthermore, although prognostic genes associated with survival outcomes have been identified in clear cell renal cell carcinoma [42,43] the prognostic impact of germline variants in ccRCC remains underexplored. This study identified novel ccRCC-specific eQTLs and eGenes by utilizing independent Japanese ccRCC cohorts and compared them with the public databases, including eQTL signals identified from GTEx healthy kidney tissues and a TCGA-KIRC cohort. These unique eQTL signals from the JP cohort should be interpreted carefully due to technical differences among the cohorts' sequencing and analysis methods. On another aspect, these differences could also be led by the population-specific variant presentations, and potential tumor tissue-specific regulations. Differences in LD structure may also contribute, but detailed LD analysis was beyond our scope. Meanwhile, significant eGene overlap across cohorts suggests conserved, housekeeping regulation. Notably, rarer variants showed larger effect sizes, underscoring the need for larger and deeper sequencing studies.

From the JP cohort, we detected significant eQTL-eGene pairs and found many of them, e.g., the top significant ones, were reported to associate with ccRCC or other cancer progressions. The lncRNA gene LINC01291 is the top eGene in the JP cohort, with the eQTL rs3771800 CC genotype resulting in a higher level of LINC01291 expression. The same regulated pattern

was identified by the GTEx kidney cohort. LINC01291 was found to increase the expression level of IGF-1R (insulin-like growth factor-1 receptor) by competing with miR-625-5p, which exhibits pro-oncogenic roles in melanoma [44]. The higher expression of IGF-1R was reported to promote cancer cell proliferation and lead to an increased risk of death for ccRCC patients [45]. In our findings, rs3771800 CC increases the LINC01291 expression, however, showing protective signals to ccRCC, which suggests LINC01291 might regulate IGF-1R differently in kidney cancer tissue. ERAP2 (endoplasmic reticulum aminopeptidase 2), the second most significant eGene in the JP cohort, encodes multifunctional enzymes that have a great biological function when cells generate major histocompatibility complex (MHC) class I binding peptides [46]. This process might influence tumor immunogenicity [47] and immune infiltration [48], in which a deficiency of ERAP2 protein can contribute to immune evasion by tumor cells [49]. Research about the anti-PD1 response in the ccRCC [50] has identified ERAP2 mutation as a signature gene related to defective antigen presentation. From our analysis, we showed that the TT genotype of rs2549782-ERAP2 decreased the gene expression level so which potentially decreased the level of enzymes. Some significant eGenes have not been reported in ccRCC research before. The gene RPS26 (Ribosomal protein S26) is a disease-related gene, and the mutation of RPS26 usually causes Diamond-Blackfan Anemia. This gene functions in pre-RNA processing, but no evidence shows its relation to kidney cancer in previous studies. Notably, the RPS26 protein was implicated in regulating the p53 response to DNA damage [51] and the p53 protein has a great impact on different cancers. The gene XRRA1 (X-Ray Radiation Resistance Associated 1) is also a gene related to DNA damage repair. A few works of literature mentioned that it might be related to cancer development, such as the study by [52] showed XRRA1 targets ATM/CHK1/2-Mediated DNA Repair in Colorectal Cancer.

By comparing with a recent published kidney-cancer GWAS [6] and TWAS [37] study, several overlapped signals were emphasized. Firstly,rs12513763 emerged as both a GWAS susceptibility loci and an eQTL identified in the JP cohort. This SNP regulates SLC6A18, a sodium cotransporter, which has not yet been reported to be associated with the prognosis of ccRCC but is highly expressed in kidney proximal tubules [53]. At the gene level, NTN4 was identified as the nearest gene to a GWAS signal and as an eGene in our JP cohort. This gene was reported in recent studies as a prognostic marker [54] and a bidirectional signaling molecule [55] for ccRCC. From the comparison with the TWAS study, we revealed 15 eGenes that were shared between our study and the TWAS findings. Notably, RCCD1, MLF2, HEATR3, and EIF1B-AS1 showed associations with both patient survival from the JP cohort and ccRCC disease risk from TWAS.

We discovered a notable number of long non-coding (lnc) RNA genes and pseudogenes among the identified eGenes. In line with our findings, an eQTL mapping analysis of the Korean Crohn's disease, pseudogenes and lncRNA genes were identified, the former of which accounted for 11% of the total eGenes [56]. Another eQTL analysis of a larger cohort of 2112 individuals revealed the roles of SNPs from the loci of pseudogenes and non-coding RNAs in the regulation of the coding genes. They have proved that pseudogenes loci with eQTL signals were more often transcribed, and some of the eQTL signals in non-coding loci were in LD with GWAS-SNPs [57]. Based on the survival analysis of the JP cohort, there are five pseudogenes and one lnc RNA gene (among the 11 eGenes) that belong to the strict replicated pairs we identified. Our survival analysis suggests that these two kinds of genes might play a role in prognosis functions. Previous studies have indicated that lncRNAs play roles in gene regulation [58,59] in cancers, and some have been identified as prognostic signatures that affect the survival of patients with cancer, including colon cancer [60] and breast cancer [61]. Specifically, for ccRCC, six lncRNAs were reported as prognostic factors and showed a high C-Index (0.853) when predicting the 5-year survival of 539 ccRCC patients [62]. It was also reported that a significant number of lncRNAs and pseudogenes were related to ccRCC prognosis by altering expression levels [63]. For example, the pseudogene PTENP1 was reported as a competing RNA that suppresses the progression of ccRCC [64] and patients with no PTENP1 expression have a lower survival rate. These findings show similarities with the prognostic analysis in the JP cohort, suggesting strong associations between the expression levels of lncRNAs and pseudogenes with ccRCC patients' prognosis. As two types of convincing regulatory factors, it's interesting that the lncRNA and pseudogenes themselves are potentially regulated by their associated eQTLs loci. The genotype of the eQTL loci differs in each individual might be the reason for the differential expression of lncRNAs and pseudogenes.

Prognostic eQTL discoveries are relatively less common compared to gene-level prognostic markers, partly due to uncertainties in how SNPs or indels impact phenotypes. The choice of inheritance model, such as additive, dominant, or recessive, can significantly influence the observed prognostic effects [65]. For instance, a similar study in multiple myeloma using additive and co-dominant models found that eQTLs for prognostic genes were associated with survival under different models [20], which partially inspired our research. In addition, although clinical parameters significantly affect prognosis, the availability and accuracy of clinical parameters may differ depending on the sample and cohort collection period. The available clinical parameters may not capture all relevant factors affecting a patient's condition, such as medication history or overall physical fitness. In general, univariate Cox analysis has been used to evaluate the associations between gene expression and outcomes, alongside adjustments for common clinicopathologic variables through separate Cox analyses [66]. In our study, we maintained both unadjusted and adjusted results to ensure that potential signals were not overlooked due to variations in clinical data collection. We compared the prognostic effects between the two cohorts by first examining the unadjusted results from the JP cohort against the unadjusted results from the TCGA cohort, followed by a similar comparison of the adjusted results from both cohorts. For SNPs, we also considered different inheritance models. All comparisons were done pairwise to ensure a thorough evaluation of both unadjusted and adjusted effects across the cohorts and under the same genetic models. As illustrated in Fig 5 and Table 3, these signals shared at least one significant signal in the same direction between the two cohorts.

In the validation phase, we discovered a total of 11 unique eGenes corresponding to 8 eQTLs. Among these eGenes, we investigated them based on their corresponding eQTL loci: TMEM254-AS1 (Chr10q22.1), DISP2 (Chr15q13.3), PWP2 (Chr21q22.3), SEPHS1P1&ERV3–1(Chr7q21.13), and CYP4F60P& ANKRD20A7P& SNX18P5 (Chr9p11.1). TMEM254-AS1 is a lncRNA transcribed from the TMEM254; its expression has been found to change in certain cancer types and is associated with survival [67,68]. DISP2 encodes a protein related to proteasome-mediated degradation and signalling by Hedgehog leads to pathological consequences in cancers, and PWP2 has been found to promote in-vitro invasion and migration of cancer cell lines. Specifically, ERV3–1 is part of human endogenous retroviruses (HERV) as an envelope protein. HERVs have shown promising diagnostic and prognostic potential in cancer, particularly in ccRCC, where their role in carcinogenesis is believed to be mediated through immunomodulatory effects [69,70]. In the results, we investigated that the ERV3–1 also somatically mutated among the JP cohort, and the eQTL rs34639489 is a missense variant that is also deleterious to protein structure. This is a relatively rare case since most of the controlling eQTLs are assigned to non-functional categories [71]. We assumed that this SNP might be associated with ERV3–1'sexpression level by affecting the mRNA translations and eternally causing variations in patients' prognosis. A group of 6 pseudogenes RP11-459O16.8, CR848007.2, GXYLT1P5, CYP4F60P, ANKRD20A7P, and SNX18P5 was regulated by the same loci (rs28612439 and rs28395168). These two SNPs and a large proportion of the prognostic eQTLs identified in the JP cohort surround this specific genomic location chromosome 9p11.1. This region falls into the centromere, which is a region with much higher gene density per centimorgan (cM), and for this reason, might be prone to be the eQTL "hotspots" [72]. Centromeres are often enriched for non-coding RNAs, pseudogenes, and repetitive elements, which may harbor regulatory activity. The centromere on chromosome 1 was identified as a putative trans-eqtl hotspot, and cancer-specific eQTLs were also enriched for heterochromatic regions in the Pan-Cancer Analysis of Whole Genomes (PCAWG) study [73]. These findings suggest that centromere regions may exhibit an open chromatin configuration [74] under some biological conditions, such as during cancer cell development. On the other hand, chromosome 9 is a frequently reported region in ccRCC-related survival studies. Most of the reports indicated that the abnormality of chromosome 9, such as deletions in the short arm 9p [75] and the monosomy of the whole chromosome 9 [76] is strongly associated with worse survival of ccRCC patients. Currently, there is no research indicating the function of these six specific pseudogenes, but there are a lot of other pseudogenes reported as factors impacting tumorigenesis, such as the PTENP1 for ccRCC as previously mentioned. The pseudogenes are usually considered as regulatory factors at the transcription level (competitive antisense RNA as inhibitors) or post-transcriptional level (competing endogenous RNA function as microRNA decoys) [77] for the expression of their parent

genes or other genes. The Pseudogene ANKRD20A7P belongs to the Ankyrin Repeat Domain 20 Family. The structure of Ankyrin Repeat domains (ANKRD) is a unique type of conservative motif that usually functions as the mediator in protein-protein interactions [78]. According to one study, a protein with the structure of ankyrin repeats, named ANKHD1, governs RCC proliferation by binding to and altering a subset of miRNAs [79]. Based on the current knowledge, we assume the expression of ANKRD20A7P might be a regulatory factor either as antisense RNA or competing endogenous RNA that affects other tumor-related gene's expression. We also suggested that genotypes of the eQTL rs28612439 regulate the expression of ANKRD20A7P, which contributes to differences in survival among patients who being diagnosed with ccRCC.

In conclusion, we developed an eQTL database derived from a Japanese cohort, offering a complementary perspective to the ccRCC research community, particularly for individuals of Asian descent. We performed survival analysis on both gene and SNP levels and found a group of eQTLs of prognostic genes is also directly prognosis. We validated our findings using the TCGA cohort, which revealed shared prognostic genetic signals and associated genes across diverse ethnic groups, despite their differing genetic backgrounds. We applied both unadjusted and adjusted survival models with different inheritance models of SNPs to prevent missing out on shared signals between two distinct cohorts. Our study highlights the regulatory roles of non-coding regions, especially pseudogenes, such as ANKRD20A7P in the progression of ccRCC. Furthermore, we linked key genes with known prognostic significance, such as ERV3–1, to specific regulatory loci, potentially aiding the development of targeted therapies. While further validation in larger cohorts is needed, these findings contribute to our understanding of the genetic basis of ccRCC progression and may help inform future studies on prognostic biomarkers in diverse populations.

This study is not free of limitations. Firstly, survival models in this study did not include the therapies that patients underwent due to limitations in the metadata. Secondly, WES data covers only a limited subset of functional regulatory regions at non-coding sites. Some significant regulatory variants might be missing in the results. The regulatory signals identified in this study have not yet been subjected to functional experiments, rather serve as a reference database for future studies. Cancer tissue transcriptomics might be affected by copy number variations (CNVs) that are generated in tumor tissue cells, which we were not able to take into account due to data limitations.

## Supporting information

**S1 Fig. Clinical characteristics of the Japanese and TCGA kidney cancer cohorts.** This includes age, gender, living days, stage at diagnosis (T), lymph node spread (N), and metastases stage (M).
(TIF)

**S2 Fig. Genomic annotation and distribution of tested variants in cis-eQTL analysis.** (A) Proportions of variants in intronic (60.9%), coding (16.3%), and regulatory (8.2%) regions. The "coding" category in the plot includes variants annotated as missense, stop gained/lost, start lost, frameshift, splice site, inframe insertion/deletion, protein-altering, transcript ablation/amplification, and feature elongation/truncation. The "regulatory" category includes variants in 5′ and 3′ untranslated regions (UTRs), promoter regions, enhancers, transcription factor binding sites, as well as upstream and downstream gene variants. The "intronic" category includes intron variants, while "intergenic" refers to variants annotated as intergenic. Variants not fitting these classifications were grouped as "other non-coding". (B) Distribution of variant distances from transcription start sites (TSSs). (C) Count distribution of variants per gene. (D) Chromosome-level density plot showing genome-wide distribution of tested variants.
(TIF)

**S3 Fig. Overview of tested gene-variant pairs in cis-eQTL mapping.** This figure shows the Manhattan plot and Q-Q plot of cis-eQTL analyses in the JP cohort. A total of 25,508 genes and 8,495,717 gene-SNP/Indel pairs were analyzed. A total of 180 genes without variant information were excluded from the analysis.
(TIF)

**S4 Fig.  Target specificity of identified eQTLs in the JP cohort.** Among 4,558 unique eQTLs, 90% target one eGene and 10% target multiple eGenes.
(TIF)

**S5 Fig.  Overlap of JP cohort eQTLs with GTEx and Pan-cancer datasets.** (A) Venn diagram showing 1,452 (31.86%) eQTLs shared with GTEx. (B) Comparison showing 2,265 (49.7%) JP eQTLs overlap with Pan-cancer ccRCC eQTLs.
(TIF)

**S6 Fig.  Somatic mutation load in the JP kidney cancer cohort.** Over 40% of samples show a log10-transformed mutation load between 2.4 and 2.6.
(TIF)

**S7 Fig.  Cox regression analysis of available clinical factors in the JP cohort.** This Figure shows the analysis results for age, gender, tumor stage (T), lymph node spread (N), and metastasis (M). Age and gender were not significantly associated with overall survival (OS), while tumor stage, lymph node spread, and metastasis showed significant associations with OS (P<0.001).
(TIF)

**S8 Fig.  Kaplan-Meier survival plots of top survival-associated eGenes and eQTLs by log-rank p-value.** These plots visualize the prognostic effects for the top 4 eGenes and eQTLs ranked by log-rank P-value. Each plot highlights the significant association between gene expression or variant status and overall survival in the JP cohort.
(TIF)

**S9 Fig.  eQTL-plots from the GTEx database, for (A)ANKRD20A7P-rs28612439 and (B) ERV3–1-rs34639489.** These two eSNPs show the same allelic effects towards gene expression patterns investigated from the JP cohort.
(TIF)

**S10 Fig.  Comparison of JP eGenes with TWAS-reported genes from a transcriptome-proteome-wide association study for RCC(PMID: 39137781).** (A) Fifteen overlapping genes were identified between JP eGenes and TWAS genes, including 11 general overlaps and 4 prognostic overlaps. (B) Distribution of these 15 overlapping genes across analysis tables. (C) Four genes overlap between TWAS genes and JP prognostic eGenes,.
(TIF)

**S1 Table.  Quality control summary of variants in the JP kidney cancer cohort.** This table shows the number of variants remaining at each QC step. A total of 284,774 variants passed QC filters, including 244,803 SNPs, 17,346 insertions, and 22,625 deletions. Of these, 283,376 variants (~99.5%) within ±1Mb windows around gene TSSs were used for eQTL analysis.
(DOCX)

**S2 Table.  List of 805 significant eGenes identified by FastQTL in the JP cohort.** eGenes were defined with FDR<0.05 from FasQTL outputs. Among these, 46 failed extra genetic models based on sensitivity analysis were excluded from survival analysis.
(XLSX)

**S3 Table.  Significant eQTL-eGene pairs identified in the JP cohort.** A total of 5,228 significant pairs, including 4,558 unique eQTL loci (FDR<0.05), were identified within the cis-window of 1Mb.
(XLSX)

**S4 Table.  List of somatic mutated genes with at least five recurrent mutations in the JP cohort.** This revealed a total of 712 genes harbor recurrent somatic mutations.
(XLSX)

**S5 Table. Prognostic eGenes identified by Cox survival analyses in the JP cohort.** Univariate Cox analysis identified 158 prognostic eGenes (p-value <0.05), while adjusted models identified 82 eGenes; 53 overlapped between models. The union set of 187 eGenes was considered as potential prognostic factors for downstream analysis.
(XLSX)

**S6 Table. Survival analysis of top eQTLs using unadjusted and adjusted Cox regression in the JP cohort.** Cox regression analysis results for 743 top eQTLs that appear in both eGenes.txt and signif_pairs.txt from the Japanese (JP) cohort. Variants with <3 Individuals per genotype were excluded. The adjusted model controls for tumor stage (T), nodal involvement (N), and metastasis (M). Hazard ratios (HR) and 95% confidence intervals (CI) were reported for each eQTL. Extreme HR values (>50) or confidence intervals with ratios >100 were converted to NA.
(XLSX)

**S7 Table. Prognostic eQTLs identified in the JP cohort (p-value <0.05).** A total of 711 significant eQTLs were identified, including 367 from unadjusted and 531 from adjusted Cox models.
(XLSX)

**S8 Table. The eGenes and eQTLs that are pairwisely associated with patient survival.** A total of 54 eGenes and 223 eQTLs forming 329 significant pairs showed consistent association with overall survival at both gene and variant levels.
(XLSX)

**S9 Table. Replicated eGene-eQTL pairs with consistent prognostic effects across cohorts.** Nineteen eGene-eQTL pairs showed consistent allelic direction and survival association in both JP and TCGA cohorts, including eight strict replicated eSNPs linked to 11 eGenes.
(XLSX)

## Acknowledgments

We acknowledge the TCGA Research network (https://www.cancer.gov/tcga) and the Japanese ccRCC cohort data provider from the University of Kyoto and the University of Tokyo. We also acknowledge the computational resource provided by the National Infrastructure for Supercomputing in Sweden (NAISS) through Uppsala Multidisciplinary Center for Advanced Computational Science (UPPMAX), under Project NAISS 2023/23–628.

## Author contributions

**Conceptualization:** Xiya Song, Cheng Zhang, Adil Mardinoglu.

**Data curation:** Xiya Song, Xiangyu Li, Meng Yuan, Yusuke Sato, Haruki Kume, Seishi Ogawa.

**Formal analysis:** Xiya Song.

**Funding acquisition:** Cheng Zhang, Adil Mardinoglu.

**Investigation:** Xiya Song.

**Methodology:** Xiya Song, Han Jin, Xiangyu Li, Meng Yuan, Hong Yang.

**Project administration:** Cheng Zhang, Adil Mardinoglu.

**Resources:** Yusuke Sato, Haruki Kume, Seishi Ogawa.

**Software:** Xiya Song.

**Supervision:** Cheng Zhang, Adil Mardinoglu.

**Visualization:** Xiya Song.

**Writing – original draft:** Xiya Song.

**Writing – review & editing:** Han Jin, Xiangyu Li, Meng Yuan, Hong Yang.

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
