## [Decision Letter · Decision Letter 0]

PGENETICS-D-25-00018

Systematically identification of survival-associated eQTLs in a Japanese kidney cancer cohort

PLOS Genetics

Dear Dr. Mardinoglu,

Thank you for submitting your manuscript to PLOS Genetics. After careful consideration, we feel that it has merit but does not fully meet PLOS Genetics's publication criteria as it currently stands. Therefore, we invite you to submit a revised version of the manuscript that addresses the points raised during the review process, including sharing the full summary statistics of the eQTL analysis.

Please submit your revised manuscript within 60 days May 03 2025 11:59PM. If you will need more time than this to complete your revisions, please reply to this message or contact the journal office at plosgenetics@plos.org. Please include the following items when submitting your revised manuscript:

We look forward to receiving your revised manuscript.

Kind regards,

John Prensner

Academic Editor

PLOS Genetics

Giorgio Sirugo

Section Editor

PLOS Genetics

Aimée Dudley

Editor-in-Chief

PLOS Genetics

Anne Goriely

Editor-in-Chief

PLOS Genetics

**Additional Editor Comments :**

We would advise the authors that publication of full summary statistics on the eQTL analysis should be shared.

**Journal Requirements:**

At this stage, the following Authors/Authors require contributions: Xiya Song, Han Jin, Xiangyu Li, Meng Yuan, Hong Yang, Yusuke Sato, Haruki Kume, Seishi Ogawa, Cheng Zhang, and Adil Mardinoglu. Please ensure that the full contributions of each author are acknowledged in the "Add/Edit/Remove Authors" section of our submission form.

The list of CRediT author contributions may be found here: https://journals.plos.org/plosgenetics/s/authorship#loc-author-contributions

https://journals.plos.org/plosgenetics/s/submission-guidelines#loc-parts-of-a-submission

**Reviewers' comments:**

Reviewer's Responses to Questions

Reviewer #1: In this study, authors applied the previously published data of whole-exome sequencing and RNA sequencing data from 100 Japanese

ccRCC patients to identify eQTLs. The prognostic effect of the expressed genes was validated using public TCGA data of 287 patients.

The advantage of this study is a series of bioinformatic methods used to do this. The disadvantage included (1) prognosis was affected by many therapies and demographic information for the patients. These factors should be all included in Cox proportional model; (2) whole-exome sequencing only includes a percentage of functional SNPs in the promoter region and/or enhancer regions. The data od eQTLs are not intact. (3) functional experiments should be helpful to enrich the evidences.

Reviewer #2: In this manuscript, Song and colleagues performed an eQTL analysis of kidney cancer tissues of a Japanese population and assessed the association of the eQTL genes/variants with cancer outcome in the same cohort. This will be a good resource for kidney cancer genetics research by adding a dataset representing East Asian populations. I have the following comments for the authors:

Major:

1. One of the main limitations of the dataset is that the genotypes are based on exome sequencing data with limited coverage beyond coding regions. The authors mentioned that variants in +/-1Mb of the TSS were used for the regression, but the coverage beyond immediate upstream of TSS or far downstream beyond transcription end site will be sparse with only coding regions of neighboring genes are covered. The authors also mentioned that “the missing genotypes were imputed by FastQTL” (line 223), but it is not clear what reference dataset and what methods were used. Moreover, it is not clear to what extent within +/-1Mb window of the TSS the variants were imputed and used for the analysis, and if imputation quality score cutoff was applied. Did the authors mean they imputed non-coding variants that were not originally covered in the WES panel? Please provide the detailed criteria and quality control in dealing with the missing genotypes within the +/-1Mb scanning window. Please also provide data on distribution of the SNPs included in eQTL analysis within this window.

2. One of the important considerations of eQTL analysis using tumor tissues is the copy number adjustment. In the regression model, the authors did not account for gene expression differences in the tumor transcriptome resulting from copy number changes in the tumor genome. If the somatic copy number information (e.g., gene-level copy number) is not available, please state this as a limitation.

3. The authors tested 805 eGenes and 4,558 cis-eQTL variants for their correlation with survival in their cohort but did not mention any multiple testing corrections for the number of genes and variants tested. Please apply proper corrections before declaring significance of prognostic eGenes/eQTLs.

4. Since this dataset could serve as a resource for the community, please share the full eQTL summary stats.

Minor:

1. Line 222: Permutation for eQTL adjusted p-value was mentioned as 100 to 1000 times but shown as 1000 to 10000 in Figure 1

2. Line247: please specify that gene expression levels (the same normalized expression levels used in the eQTL study?) was used for the association between eGenes and overall survival to clarify the analysis for the readers.

3. Line 258: please clarify whether both the HR and eQTL slope are relative to the minor allele. FastQTL pipeline typically generates slope relative to alternative allele, which might not be the same as the minor allele in this population. Aligning eQTL and survival analysis to the same allele is important given the authors later filter results based on consistency of the direction between eQTL and eGenes.

4. Line 343: “Among the 518 eGenes”. Shouldn’t it be 805 eGenes?

5. Line 352-354: The authors cannot attribute 50% of eQTL identified in this dataset as potential “ethnic specific” eQTLs without proper assessment of allele frequencies, for example. There could be multiple other reasons, including differences in the genotype and expression detection methods, imputation methods, tumor histology, grade, somatic drivers, eQTL analysis methods, to name a few.

Reviewer #3: The manuscript by Song et al reports a moderate yet comprehensive genomic-transcriptomic analysis of 100 Japanese clear cell renal cell carcinoma patients. The authors perform germline eQTL discovery, somatic mutation calling as well as a novel and interesting association analysis with survival. Albeit the moderate size of the cohort, the resource can be extremely valuable to study the differences in genetics ccRCC in different populations. I have the following comments:

• Ref 6, Please also include the most recent larger GWAS of ccRCC (PMID: 38671320).

• Fig 2B: Given the overall lower sample size of the JP cohort, some of the top results might be due to few samples in one genotype group. E.g. For RPS26 CC has two samples(?). So I would encourage the authors to do a further sensitivity analyses for the top findings using dominant/recessive models for each allele to weed out any possibility of low-sample bias results.

• Line 325: Please quantify/mention the “small proportion” of eQTL with more than one gene. Please use quantifications whenever possible rather than using qualitative descriptors.

• Fig 3. I would suggest adding a plot describing the relation between effect allele frequency and effect size which can inform the extent of selection. In Fig 3C the coloring of the density plots overlap which makes the “all eQTL” plot difficult to see. Fig 3D: “Lorem ipsum”?

• For Fig 3B-C, given that this is a WES study, how is the major portion of variants intron variants and substantial effects exist in the regulatory regions?

• Line 349: Is there a typo in the p-value? In general, I am not sure about how precise the hypergeometric test is when testing for overlap of significant eQTLs which are strongly correlated.

• Line 352: The population specificity might be due to a plethora of reasons: (a) the variant is absent in one population (b) the effect allele frequencies differ (c) the LD structures differ (d) the eQTLs indeed have different effects. To me the most interesting scenarios are (a) and (d). Can the authors comment on these?

• The discussion on section 3.3 should be represented by a somatic mutation figure.

• Line 422: Typo: “derived”?

• I would commend the authors for nicely hypothesizing and summarizing the integration of eQTLs in survival analysis.

• Although the replication of survival association in TCGA is limited, I would recommend the authors perform a slightly lenient search in terms of similar direction of effects. I would also strongly recommend the authors compare the results with the TWAS reported in PMID: 39137781, which can give a complementary evidence.

• Fig 6. Direct screenshots from GTEx server is not recommended to be included as a main figure. This can go in as text and/or supplementary as it is not the work of the authors directly.

**Have all data underlying the figures and results presented in the manuscript been provided?**

Reviewer #1: Yes

Reviewer #2: **No: ** Full summary stats of the eQTL analysis were not shared, and I recommend authors to make them public

Reviewer #3: None

PLOS authors have the option to publish the peer review history of their article (what does this mean? ). If published, this will include your full peer review and any attached files.

**Do you want your identity to be public for this peer review?** For information about this choice, including consent withdrawal, please see our Privacy Policy .

Reviewer #1: **Yes: ** Guangwen Cao

Reviewer #2: No

Reviewer #3: No

**Figure resubmission:**
---

## [Decision Letter · Decision Letter 1]

PGENETICS-D-25-00018R1

Systematically identification of survival-associated eQTLs in a Japanese kidney cancer cohort

PLOS Genetics

Dear Dr. Mardinoglu,

Thank you for submitting your manuscript to PLOS Genetics. After careful consideration, we feel that it has merit but does not fully meet PLOS Genetics's publication criteria as it currently stands. Therefore, we invite you to submit a revised version of the manuscript that addresses the points raised during the review process.  Please address the comments made by Reviewer #1 in your revised manuscript.

Please submit your revised manuscript within 30 days Jun 28 2025 11:59PM. If you will need more time than this to complete your revisions, please reply to this message or contact the journal office at plosgenetics@plos.org. Please include the following items when submitting your revised manuscript:

We look forward to receiving your revised manuscript.

Kind regards,

John Prensner

Academic Editor

PLOS Genetics

Giorgio Sirugo

Section Editor

PLOS Genetics

Aimée Dudley

Editor-in-Chief

PLOS Genetics

Anne Goriely

Editor-in-Chief

PLOS Genetics

**Reviewers' comments:**

Reviewer's Responses to Questions

**Comments to the Authors:**

Reviewer #1: This study systematically identifies survival-associated expression quantitative trait loci (eQTLs) in a Japanese clear cell renal cell carcinoma (ccRCC) cohort and validates findings in the TCGA-KIRC dataset. The work is well-designed, methodologically sound, and provides novel insights into the prognostic role of germline regulatory variants in ccRCC, particularly in an understudied Asian population. However, several issues require clarification and improvement to strengthen the manuscript’s impact and reliability.

1. Caution is needed when attributing unique eQTLs to ancestry (e.g., "50% are Asian-specific"). Differences could stem from technical factors (e.g., WES coverage) or tumor-specific regulation.

2. Report hazard ratios (HRs) and confidence intervals for top eQTLs to assess clinical relevance.

3. Fig 6A: Label axes with effect directions (e.g., "Protective" vs. "Risk" alleles).

4. Ensure all cited tables/figures (e.g., somatic mutations in Suppl. Table 4) are referenced in the main text.

5. In discussion: Compare findings to recent ccRCC GWAS (e.g., PMID: 38671320) and TWAS (e.g., PMID: 39137781).

6. Discuss why chromosome 9p11.1 emerges as an eQTL hotspot (e.g., open chromatin, high gene density).

The manuscript is well-written and analytically thorough but requires clarifications on statistical methods, biological context, and presentation. No additional data is needed.

Reviewer #2: The authors addressed most of the points raised by this referee.

Minor comments:

Supplementary Fig2A: please specify what the "regulatory" category includes in the legend

Table 2: I suggest that the authors use the term "significant" instead of "solid" to describe their findings.

Reviewer #3: The authors have addressed all my queries satisfactorily. I congratulate them on this important work.

**Have all data underlying the figures and results presented in the manuscript been provided?**

Reviewer #1: Yes

Reviewer #2: Yes

Reviewer #3: None

PLOS authors have the option to publish the peer review history of their article (what does this mean? ). If published, this will include your full peer review and any attached files.

**Do you want your identity to be public for this peer review?** For information about this choice, including consent withdrawal, please see our Privacy Policy .

Reviewer #1: No

Reviewer #2: No

Reviewer #3: No

**Figure resubmission:**
---

## [Editor Report · Decision Letter 2]

PGENETICS-D-25-00018R2

Systematically identification of survival-associated eQTLs in a Japanese kidney cancer cohort

PLOS Genetics

Dear Dr. Mardinoglu,

Thank you for submitting your manuscript to PLOS Genetics. After careful consideration, we feel that it has merit but does not fully meet PLOS Genetics's publication criteria as it currently stands. Therefore, we invite you to submit a revised version of the manuscript that addresses the points raised during the review process.

Please submit your revised manuscript within 30 days Jul 11 2025 11:59PM. If you will need more time than this to complete your revisions, please reply to this message or contact the journal office at plosgenetics@plos.org. Please include the following items when submitting your revised manuscript:

We look forward to receiving your revised manuscript.

Kind regards,

John Prensner

Academic Editor

PLOS Genetics

Giorgio Sirugo

Section Editor

PLOS Genetics

Aimée Dudley

Editor-in-Chief

PLOS Genetics

Anne Goriely

Editor-in-Chief

PLOS Genetics

**Additional Editor Comments:**

Please adjust the manuscript text to satisfy the suggestions by Reviewer #1

**Reviewers' comments:**

**Figure resubmission:**
---

## [Editor Report · Decision Letter 3]

Dear Dr Mardinoglu,

We are pleased to inform you that your manuscript entitled "Systematically identification of survival-associated eQTLs in a Japanese kidney cancer cohort" has been editorially accepted for publication in PLOS Genetics. Congratulations!

Yours sincerely,

John Prensner

Academic Editor

PLOS Genetics

Giorgio Sirugo

Section Editor

PLOS Genetics

Aimée Dudley

Editor-in-Chief

PLOS Genetics

Anne Goriely

Editor-in-Chief

PLOS Genetics

Comments from the reviewers (if applicable):

**Data Deposition**

http://datadryad.org/submit?journalID=pgenetics&manu=PGENETICS-D-25-00018R3

**Press Queries**

---

## [Editor Report · Acceptance letter]

PGENETICS-D-25-00018R3

Systematically identification of survival-associated eQTLs in a Japanese kidney cancer cohort

Dear Dr Mardinoglu,

We are pleased to inform you that your manuscript entitled "Systematically identification of survival-associated eQTLs in a Japanese kidney cancer cohort" has been formally accepted for publication in PLOS Genetics! Your manuscript is now with our production department and you will be notified of the publication date in due course.

With kind regards,

Olena Szabo

PLOS Genetics

On behalf of:
